# Using snowflake surface-area-to-volume ratio to model and interpret snowfall triple-frequency radar signatures

Mathias Gergely[1], Steven J. Cooper[1], and Timothy J. Garrett[1]

[1]Department of Atmospheric Sciences, University of Utah, 135 S 1460 E Room 819, Salt Lake City, UT 84112, USA.

*Correspondence to:* Mathias Gergely (mathias.gergely@utah.edu)

**Abstract.** The snowflake microstructure determines the microwave scattering properties of individual snowflakes and has a strong impact on snowfall radar signatures. In this study, individual snowflakes are represented by collections of randomly distributed ice spheres where the size and number of the constituent ice spheres are specified by the snowflake mass and surface-area-to-volume ratio (SAV) and the bounding volume of each ice sphere collection is given by the snowflake maximum dimension. Radar backscatter cross sections for the ice sphere collections are calculated at X-, Ku-, Ka-, and W-band frequencies and then used to model triple-frequency radar signatures for exponential snowflake size distributions (SSDs). Additionally, snowflake complexity values obtained from high-resolution multi-view snowflake images are used as an indicator of snowflake SAV to derive snowfall triple-frequency radar signatures. The modeled snowflake triple-frequency radar signatures cover a wide range of triple-frequency signatures that were previously determined from radar reflectivity measurements and illustrate characteristic differences related to snow type, quantified through snowflake SAV, and snowflake size. The results show high sensitivity to snowflake SAV and SSD maximum size but are generally less affected by uncertainties in the parameterization of snowflake mass, indicating the importance of snowflake SAV for the interpretation of snowfall triple-frequency radar signatures.

## 1 Introduction

Snowfall retrievals from radar remote sensing of snow clouds are highly sensitive to the applied characterization of the snowflake microstructure, i.e., of snowflake mass and shape (e.g., Matrosov, 2007; Liu, 2008; Kulie et al., 2010; Cooper et al., 2017). To analyze and model snowfall radar signatures, snowflakes have often been represented by (i) mixed ice–air spheres or spheroids parameterized with respect to snowflake size and aspect ratio (e.g., Matrosov, 1992; Hogan et al., 2006, 2012) or by (ii) detailed three-dimensional (3D) shape models of single snow crystals or aggregate snowflakes based on various idealized ice crystals like bullet rosettes, dendrites, plates, or columns (e.g., Kulie and Bennartz, 2009; Nowell et al., 2013; Ori et al., 2014; Honeyager et al., 2016).

In recent years, several studies have found that the 'soft' spheroidal particle model, where the volume, density, and complex index of refraction of a homogeneously mixed ice–air spheroid are derived from the snowflake size, mass, and aspect ratio, yields a realistic description of microwave backscatter only for small snowflakes and at low frequencies (e.g., Petty and Huang, 2010; Tyynelä et al., 2011; Nowell et al., 2013). Furthermore, the analysis of radar reflectivity measurements collected simul-

taneously at three microwave frequency bands has shown that the range of observed snowfall triple-frequency radar signatures is much larger than the total range of modeled snowfall radar signatures when representing snowflakes by soft spheroids; especially triple-frequency radar signatures of snowfall characterized by large aggregate snowflakes fall outside the modeled range (Leinonen et al., 2012; Kulie et al., 2014; Kneifel et al., 2015). Using detailed 3D shape models instead of soft spheroids leads to a wider range of modeled snowfall triple-frequency radar signatures and indicates better agreement between observed and modeled snowfall radar signatures.

Due to the large variety of (visually distinct) snow types defined by characteristic geometric shapes resembling the snowflake microstructure, such as planar dendrites or aggregates of plates (Magono and Lee, 1966; Kikuchi et al., 2013; Fontaine et al., 2014), and the high natural variability of snowflake microstructural properties like size and aspect ratio (e.g., Brandes et al., 2007; Gergely and Garrett, 2016), modeling microwave backscatter in snowfall based on detailed snowflake 3D shape models requires significant computational resources and time, e.g., when determining backscatter cross sections for a large number of snowflake models with the widely used discrete dipole approximation (Draine and Flatau, 1994). Therefore, it would be desirable to identify 'effective' microstructural parameters that quantify snowflake shape independent of snow type and still explain important features of observed and modeled snowfall radar signatures, thus further constraining snowflake shape for snowfall remote sensing.

In materials science, four basic characteristics play a central role for an objective and quantitative description of 3D microstructures: volume fraction or equivalently (mass) density, surface area per volume, integrated mean curvature per volume, and integrated Gaussian curvature per volume (Ohser and Mücklich, 2000). Physical and chemical properties strongly depend on these characteristics and can often already be analyzed faithfully when the 3D microstructure is quantified through all or some of these four characteristics. Ice volume fraction or snow density and the ratio of ice surface area to volume are crucial for modeling light scattering and radiative transfer at optical wavelengths in falling and deposited snow, for example (Grenfell and Warren, 1999; Grenfell et al., 2005; Kokhanovsky and Zege, 2004; Picard et al., 2009; Gergely et al., 2010). Besides snowflake density, however, none of these four basic characteristics have been investigated to evaluate the impact of snowflake shape on snowfall microwave scattering signatures.

In this study, snowflake density and surface-area-to-volume ratio (SAV) are used to model snowflake backscatter cross sections at X-, Ku-, Ka-, and W-band frequencies and then derive snowfall triple-frequency radar signatures for realistic snowflake size distributions. The impact of snowflake SAV on snowfall triple-frequency radar signatures is analyzed based on high-resolution snowflake imaging data collected with the Multi-Angle Snowflake Camera (MASC; Garrett et al., 2012), a pre-established density–diameter relationship for deriving snowflake mass from snowflake maximum dimension according to Heymsfield et al. (2004), and the snowflake SAV range given by Honeyager et al. (2014).

First, MASC measurements are presented in Sect. 2. The applied method for modeling snowflake backscatter cross sections and snowfall triple-frequency radar signatures is described in Sect. 3. Individual snowflakes are represented by collections of ice spheres where the size and number of the constituent ice spheres are specified by the snowflake mass and SAV and the bounding volume of each ice sphere collection is defined by the snowflake maximum dimension. Backscatter cross sections of these collections of ice spheres are calculated with the generalized multiparticle Mie solution (Xu, 1995; Xu and Å. S. Gustafson,

2001). For the same snowflake mass, different SAV values lead to collections of ice spheres characterized by a different ice sphere size and number. This characteristic forms the basis for analyzing the impact of snowflake SAV on modeled snowflake backscatter cross sections and snowfall triple-frequency radar signatures in Sect. 4. The analysis includes a comparison with snowfall triple-frequency radar signatures determined for soft spheroids and for snowflakes modeled according to the self-similar Rayleigh–Gans approximation (Hogan and Westbrook, 2014; Hogan et al., 2017). Additionally, snowflake complexity values obtained from MASC images are used as an indicator of snowflake SAV to derive snowfall triple-frequency radar signatures. The results are discussed in the context of observed and modeled snowfall radar signatures that were presented in previous studies. Section 5 summarizes the findings and conclusions.

## 2  Snowflake observations

First, the Multi-Angle Snowflake Camera (MASC) and the derived snowflake microstructural properties are described briefly (a more detailed description of a similar MASC model using slightly different camera optics and of the performed MASC image analysis was given by Garrett et al., 2012). As the applied approach for modeling the impact of snowflake SAV on snowfall radar signatures is partly based on collected snowflake data, MASC measurement results are also presented before the modeling method is introduced in Sect. 3.

### 2.1  Measurement method

Estimates of near-surface snowflake microstructural properties are obtained from MASC photographs taken at Alta (UT, USA) and at Barrow (AK, USA) during winter 2013–2014 and spring 2014. The MASC provides multi-view snowflake images from three cameras that are separated by $36°$ and point at an identical focal point at a distance of 10 cm. Snowflake images are recorded at a resolution of about 30 μm with horizontal fields of view of about 40 mm at the focal-point distance. The cameras and three light-emitting diodes serving as flash lights are triggered simultaneously at a maximum rate of 2 Hz as snowflakes fall through an array of near-infrared emitter–detector pairs sampling the horizontal fields of view of the cameras. Snowflakes with maximum dimensions of 0.2 mm and larger are recorded by the MASC and identified in the images using a Sobel edge detection algorithm. Figure 1 shows images of two snowflakes captured by the MASC center camera at Alta.

In this study, MASC images are used to derive the snowflake diameter $D$ or maximum dimension along the snowflake major axis, the orientation angle $\theta$ of the snowflake major axis with respect to the horizontal plane, and the snowflake complexity $\chi$ defined as the ratio of the snowflake perimeter to the circumference of a circle with the same area as the snowflake projection image (illustrated in Fig. 1). For all snowflakes, $D$, $\theta$, and $\chi$ are given as average values determined from the MASC single-view images of the snowflakes.

The applied definition of $\chi$ quantifies snowflake complexity based on the boundary curve length of two-dimensional (2D) snowflake images. Projection images of spherical snow particles are characterized by a circular boundary curve independent of viewing direction, and thus by a complexity of $\chi = 1$. As a circle has the shortest perimeter of any boundary curve for a given enclosed area, all non-spherical particle shapes lead to complexity values of $\chi > 1$. Accordingly, heavily rimed graupel snow

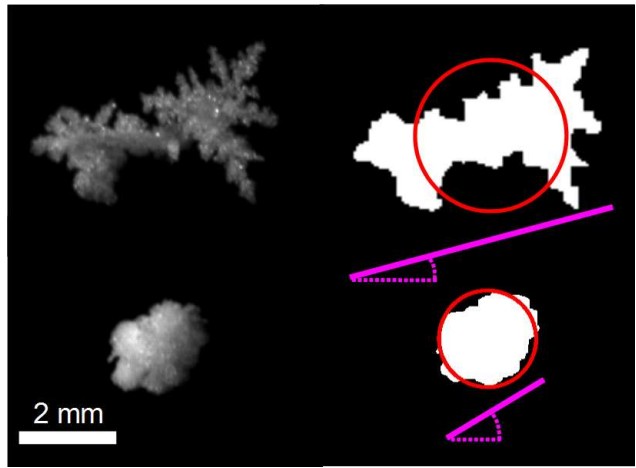

**Figure 1.**

is described by a low snowflake complexity of $\chi \approx 1$ and large aggregate snowflakes are characterized by higher complexity values (see examples in Fig. 1). Lower $\chi$ values are then expected to indicate stronger snowflake riming in general (see also Garrett and Yuter (2014) who used a definition of snowflake complexity which additionally included brightness variations within each MASC image to classify snowflakes according to their degree of riming).

One MASC was installed at Alta Ski Resort at 2590 m above sea level (a.s.l) in Collins Gulch within the Wasatch Mountain Range. A second MASC was located at Barrow at the North Slope of Alaska Atmospheric Radiation Measurement (ARM) site at 10 m a.s.l., approximately 500 km north of the Arctic Circle on the coast of the Arctic Ocean.

## 2.2    Measurement results

Figure 2 shows the distributions of snowflake diameter $D$, complexity $\chi$, and orientation angle $\theta$ derived from all qualifying
MASC observations with realistic complexity values of $\chi \geq 1$ that were collected at Alta from December 2013 to April 2014 and at Barrow in April and May 2014, resulting in a MASC data set of $4.4 \cdot 10^5$ sampled snowflakes. Snowflake size distributions $N(D)$ are expressed as frequency size distributions and reflect the number of snowflakes sampled at Alta ($4.3 \cdot 10^5$) and at Barrow ($10^4$). For snowflake complexity and orientation, the presented relative distributions are normalized with respect to the maximum values $N_{\max}(\chi)$ and $N_{\max}(\theta)$ of the respective frequency distributions $N(\chi)$ and $N(\theta)$.

The distributions of snowflake diameters and complexities in Fig. 2 are dominated by small values and show exponential decay for diameters of $D \gtrsim 1$ mm and for the entire complexity range of $\chi \geq 1$. In contrast to snowflake diameters and complexities, snowflake orientation angles are characterized by a nearly uniform distribution with mean values of $\overline{\theta} = 40\,^\circ$ derived for the set of MASC observations at Alta and $\overline{\theta} = 45\,^\circ$ at Barrow.

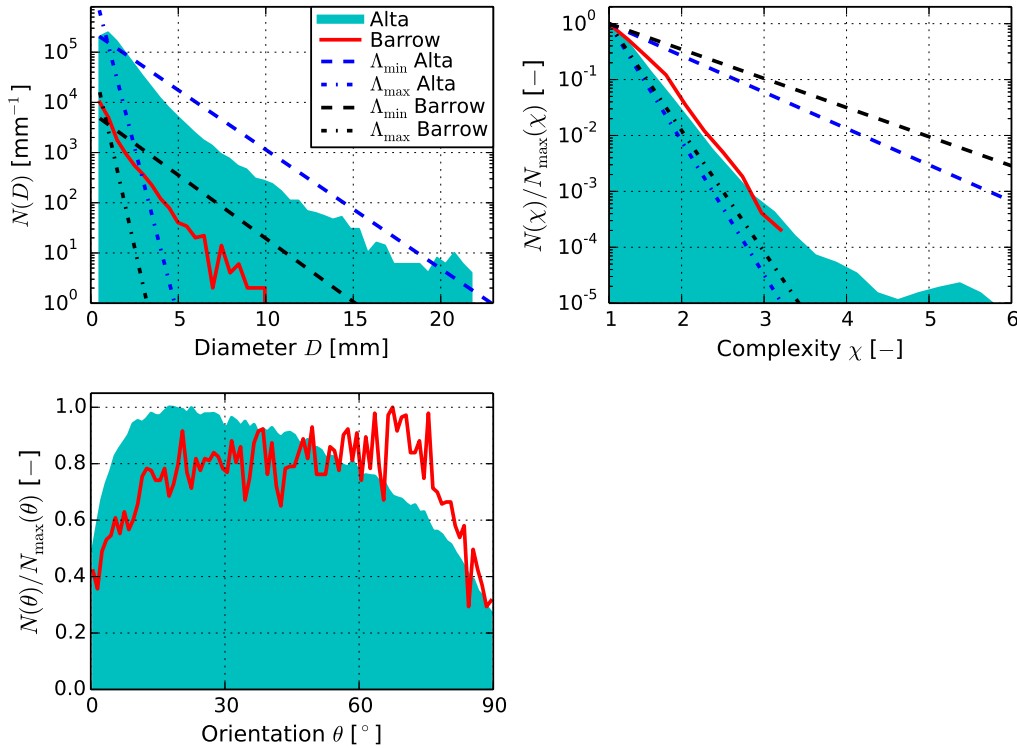

**Figure 2.**

Similar to previous studies that have used exponential snowflake size distributions to describe snowfall (e.g., Matrosov, 2007; Kneifel et al., 2011), snowflake (frequency) size distributions $N(D)$ [mm$^{-1}$] in this study are expressed through

$$N(D) = N_0 \exp(-\Lambda D) \,, \tag{1}$$

where $\Lambda$ is the exponential slope parameter specifying the width of the distribution and $N_0$ [mm$^{-1}$] denotes the scaling factor
5   determined by the snowflake sample size. Commonly, $N(D)$ and $N_0$ are additionally normalized with respect to atmospheric volume to account for the atmospheric snow water content, giving $N(D)$ and $N_0$ in units of mm$^{-1}$ m$^{-3}$. As the normalization of $N(D)$ has no impact on the analyzed dual-wavelength ratios of modeled $Z_e$ in Sect. 4.2, the scaling factor $N_0$ is ignored in the analysis and exponential distributions are specified only through the exponential slope parameter $\Lambda$.

Exponential snowflake size distributions $N(D)$ were fitted to MASC data restricted to $D > 1$ mm and collected for 47
10   snowstorms at Alta and 7 snowstorms at Barrow. These snowstorms lasted between 4 h and 24 h, and $10^2$ to $10^4$ snowflakes were recorded during each snowstorm. Small sample sizes of $10^2$ snowflakes correspond to snowstorms at Barrow marked by very low snowfall liquid equivalent of less than about 1 mm and by strong crosswinds that affected overall sampling efficiency. Large sample sizes of up to $10^4$ snowflakes correspond to high-intensity snowfall at Alta. For each snowstorm, the sampled

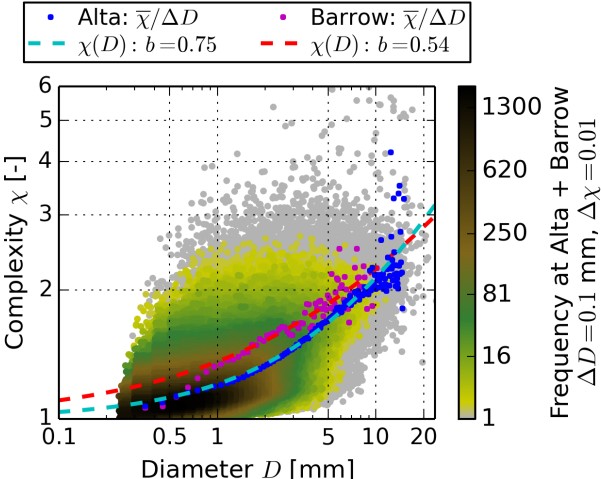

**Figure 3.**

snowflakes were divided into 20 size bins according to their diameter $D$. An exponential snowflake size distribution $N(D)$ was then determined by the non-linear least squares method for fitting Eq. (1) to the binned snowflake size distribution.

For uniform visualization in Fig. 2, $N(D)$ curves illustrating the total range of exponential size distributions fitted to the MASC data from Alta and from Barrow were rescaled to the total number of snowflakes sampled at the respective location.

At Alta, $N(D)$ are characterized by exponential slope parameters of $\Lambda_{\min} = 0.5$ mm$^{-1} \leq \Lambda \leq \Lambda_{\max} = 3.1$ mm$^{-1}$ with mean $\overline{\Lambda} = 1.2$ mm$^{-1}$. At Barrow, the range of $N(D)$ is given by $0.6 \leq \Lambda \leq 3.6$ mm$^{-1}$ with mean $\overline{\Lambda} = 1.5$ mm$^{-1}$. The derived exponential slope parameters yield snowflake size distributions $N(D)$ that are in line with previously presented snowflake size distributions using different measurement methods, e.g., by Brandes et al. (2007) and Tiira et al. (2016), with their reported median volume diameters $D_0$ of the derived snowflake size distributions converted to $\Lambda = 3.67/D_0$ for $N(D)$ given by Eq. (1).

For each analyzed snowstorm, the sampled snowflakes were also divided into 20 bins according to their complexity $\chi$, and an exponential snowflake complexity distribution $N(\chi) = N_1 \exp(-\Lambda\chi)$ was fitted to the binned distribution by the non-linear least squares method. At Alta, the range of $N(\chi)$ is characterized by exponential slope parameters of $\Lambda_{\min} = 1.5 \leq \Lambda \leq \Lambda_{\max} = 5.5$ with mean $\overline{\Lambda} = 3.3$. At Barrow, a range of $1.2 \leq \Lambda \leq 5.0$ is found with mean $\overline{\Lambda} = 2.2$ (see Fig. 2).

To illustrate the correlation between snowflake diameter $D$ and complexity $\chi$, Fig. 3 shows a logarithmic 2D histogram

of the frequency distributions for $D$ and $\chi$ at Alta and Barrow (Fig. S1 in the Supplement shows the corresponding non-logarithmic 2D histogram). Calculated mean complexity values $\overline{\chi}$ per size bin $\Delta D$ are shown separately for both MASC data sets collected at Alta ($\overline{\chi}/\Delta D$ given for $D \leq 15$ mm) and at Barrow ($\overline{\chi}/\Delta D$ given for $D \leq 10$ mm) to indicate typical snowflake complexities at the two locations. Despite the skewed distribution of $\chi$ within the size bins, the choice of whether typical snowflake complexities are quantified through the mean or through the median complexity per size bin has only a minor

influence on the derived results in this study and does not affect the drawn conclusions.

As already seen in Fig. 2, small values of $D$ and $\chi$ dominate the distributions in Fig. 3. Additionally, mean complexity $\overline{\chi}$ generally increases with increasing snowflake diameter. Notably, snowflake complexities of $\chi = 1.0$ are not observed for snowflake diameters of $D \gtrsim 3$ mm. These results are consistent with previous observations suggesting that larger snowflakes are generally aggregates characterized by a high complexity of the snowflake microstructure (Garrett and Yuter, 2014).

Based on the mean snowflake complexity values $\overline{\chi}$ per size bin $\Delta D$ shown in Fig. 3, a snowflake complexity–diameter relationship is then defined by a modified power law as

$$\chi(D) = 1 + aD^b \,, \tag{2}$$

with parameters $a$ and $b$. Power laws have been applied to parameterize a variety of snowflake properties $y$ with respect to snowflake size, illustrated by the density–diameter relationship in Eq. (3), for example. In Eq. (2), the constant of $\chi_0 = 1$ is
added to the commonly used pure power law of the form $y(D) = aD^b$ due to the definition of $\chi$, which leads to a minimum value of $\chi_{\min}(D) = \chi_0 = 1$ (Sect. 2.1).

Figure 3 shows the two $\chi(D)$ curves for the MASC data from Alta and from Barrow determined by the non-linear least squares method for fitting Eq. (2) to the mean complexity values $\overline{\chi}$ per size bin $\Delta D$. These two $\chi(D)$ relationships, with fitted parameters of $a = 0.20, b = 0.75$ at Alta and $a = 0.36, b = 0.54$ at Barrow, are dominated by the power-law term of $aD^b$ for
large snowflakes and thus follow the observed increase in $\chi$ with increasing snowflake diameter, but also reflect the observed convergence of $\chi \to 1$ for small snowflakes. Furthermore, the mean complexity values $\overline{\chi}$ per size bin $\Delta D$ and the two derived $\chi(D)$ curves generally indicate lower snowflake complexities (at a given snowflake diameter) for the MASC data recorded at Alta.

## 3    Modeling method

In this study, snowflakes are specified by their diameter, mass, and surface-area-to-volume ratio (SAV). Snowflake diameters were derived from a large set of MASC observations in Sect. 2. In Sect. 3, the quantification of snowflake mass and SAV is described, and the approach for modeling snowflake backscatter cross sections and snowfall triple-frequency radar signatures is presented.

### 3.1    Snowflake mass

No coincident measurements of snowflake mass are available for the analyzed MASC data in Sect. 2. Therefore, snowflake mass is derived from measured snowflake diameter $D$ following a previously determined density–diameter relationship that uses a similar definition of snowflake diameter (Heymsfield et al., 2004, abbreviated as 'H04' throughout the text). H04 determined effective ice-cloud particle densities by combining observations by airborne 2D optical array probes with coincident measurements of cloud ice water content. According to their results, snowflake density $\rho_{\mathrm{f}}$ [g cm$^{-3}$] and mass $m_{\mathrm{f}}$ [mg] are
calculated from snowflake maximum dimension $D$ [mm] for a spherical snowflake bounding volume $V_{\mathrm{f}}$ of diameter $D$:

$$\rho_{\mathrm{f}}(D) = 0.104 D^{-0.950} \tag{3}$$

and

$$m_{\mathrm{f}}(D) = \rho_{\mathrm{f}}(D)V_{\mathrm{f}} = \frac{\pi}{6}\rho_{\mathrm{f}}(D)D^3 \ . \tag{4}$$

Here, derived $\rho_{\mathrm{f}}(D)$ values are limited to the density of pure ice $\rho_{\mathrm{ice}} = 0.917 \ \mathrm{g \ cm^{-3}}$, leading to densities of $\rho_{\mathrm{f}}(D) = \rho_{\mathrm{ice}}$ for snowflakes with $D \leq 0.1$ mm.

With Eqs. (3) and (4), snowflake mass $m_{\mathrm{f}}$ can alternatively be expressed through the radius $r_{\mathrm{eq}}$ of a single mass-equivalent ice sphere given by

$$r_{\mathrm{eq}}^3(D) = \frac{3m_{\mathrm{f}}(D)}{4\pi\rho_{\mathrm{ice}}} \ . \tag{5}$$

     Analyzed snowflake and snowfall backscatter properties in Sect. 4 are determined from different modeling approaches that all rely on the same parameterization of snowflake mass following Eqs. (3)–(5). The impact of the parameterization of

snowflake mass on the presented results and conclusions is evaluated by uniformly increasing and decreasing all snowflake densities $\rho_{\mathrm{f}}(D)$ obtained from Eq. (3) by 25 % and by 50 %.

### 3.2   Snowflake surface-area-to-volume ratio

The normalized snowflake surface-area-to-volume ratio $\xi$ is defined as the ratio of snowflake surface-area-to-volume ratio $\mathsf{SAV}_{\mathrm{f}}$ to the surface-area-to-volume ratio $\mathsf{SAV}_{\mathrm{s}}$ of a mass-equivalent ice sphere:

$$\xi = \frac{\mathsf{SAV}_{\mathrm{f}}}{\mathsf{SAV}_{\mathrm{s}}} \ . \tag{6}$$

This definition of $\xi$ for snowflake 3D microstructures is conceptually similar to the definition of the complexity $\chi$ applied to snowflake 2D projection images in Sect. 2. For a given ice volume or mass, an ice sphere has the smallest surface area of any 3D microstructure and a surface-area-to-volume ratio of $\mathsf{SAV}_{\mathrm{f}} = \mathsf{SAV}_{\mathrm{s}} = 3/r$ with ice sphere radius $r$, leading to a normalized SAV of $\xi = 1$. Increasing values of $\xi > 1$ then imply a larger deviation of the snowflake shape from an ice sphere, and thus an

increasing complexity of the snowflake 3D microstructure.

     Snowflake SAV is quantified from the total range of $\xi$ values determined by Honeyager et al. (2014). They used a Voronoi cell-based approach to define an effective SAV by Eq. (6) for their database of snowflake 3D shape models and found values of $1 \leq \xi \leq 5$.

     The impact of snowflake SAV on snowfall radar signatures is analyzed based on synthetically generated expressions $\xi(D)$.

These $\xi(D)$ relate normalized SAV to snowflake diameter with $1 \leq \xi(D) \leq 5$ for $0 \leq D \leq D_{\mathrm{max}}$, where $D_{\mathrm{max}}$ refers to the maximum diameter of the snowflake size distribution. Based on the MASC observations in Sect. 2.2 where the average snowflake complexity $\chi(D)$ for all snowflakes with diameter $D$ was derived from snowflake 2D projection images and expressed through a power law plus constant of one in Eq. (2), $\xi(D)$ relationships indicating the complexity of the snowflake 3D microstructure are again formulated as modified power laws of

$$\xi(D) = 1 + pD^q \ . \tag{7}$$

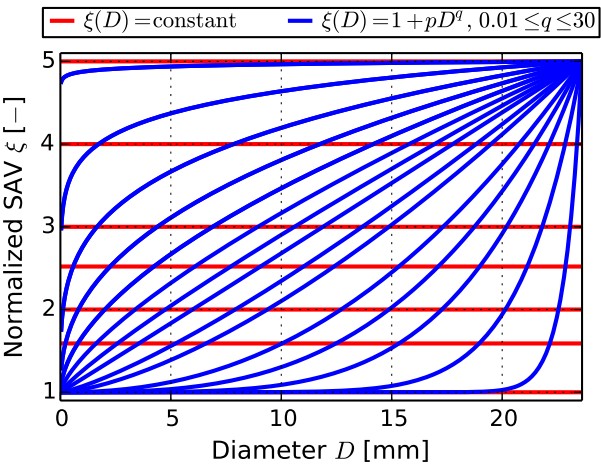

**Figure 4.**

Figure 4 shows several $\xi(D)$ curves that illustrate the total range of power-law exponents $q$ considered in the analysis, including constant values determined by setting $q = 0$. The parameter $p$ is merely a scaling factor confining Eq. (7) to the interval of $1 \leq \xi(D) \leq 5$. Only monotonically increasing $\xi(D)$ with $q \geq 0$ are considered because the analyzed MASC observations in Sect. 2.2 indicated an overall increase in snowflake complexity with increasing snowflake size.

Constant $\xi(D) = 1.0, 1.6, 2.0, 2.5, 3.0, 4.0$, and $5.0$ shown in Fig. 4 are used to model snowflake backscatter cross sections in Sect. 4.1 and lead to a wide range of snowfall triple-frequency radar signatures in Sect. 4.2. The discussion of how snowflake surface-area-to-volume ratio affects modeled snowfall triple-frequency radar signatures in Sect. 4.2 focuses on these constant $\xi(D)$. Nonetheless, non-constant $\xi(D)$ given by Eq. (7) with exponents $q > 0$ are included in the analysis to outline the total range of modeled snowfall triple-frequency radar signatures and to establish a relationship between normalized snowflake

surface-area-to-volume ratio $\xi$ and snowflake complexity $\chi$ that reflects the similarity of these two characteristics and can be applied to estimate $\xi(D)$ relationships for the recorded MASC data at Alta and at Barrow.

     The method for relating $\xi$ to $\chi$ uses the two complexity–diameter relationships $\chi(D)$ fitted to the MASC data in Fig. 3. To estimate $\xi(D)$ relationships at Alta and at Barrow, it is assumed that the snowflake complexity range of $1 \leq \chi(D) \leq \chi(D_{\mathrm{max}}) = \chi_{\mathrm{max}}$ at each location corresponds to the full snowflake SAV range of $1 \leq \xi \leq 5$ with

$$\xi(\chi) = 1 + \frac{5-1}{\chi_{\mathrm{max}} - 1}(\chi(D) - 1) \,. \tag{8}$$

After inserting Eq. (2) for $\chi(D)$, Eq. (8) leads to a modified power law for $\xi(D)$ given by Eq. (7), with power-law exponent of $q = b$. Only the scaling factor $a$ in Eq. (2) is modified by Eq. (8) to map $\chi(D)$ onto the interval of $1 \leq \xi \leq 5$.

     High values of $q \gg 1$ in Eq. (7) lead to $\xi(D)$ relationships marked by a steep increase from $\xi = 1$ to $\xi = 5$ for large snowflake diameters (see Fig. 4), corresponding to a sudden change in snowflake shape from ice spheres to more complex 3D microstruc-

tures. This is an unrealistic description of snowflake shape because such an abrupt transition is not seen in snowflake observa-

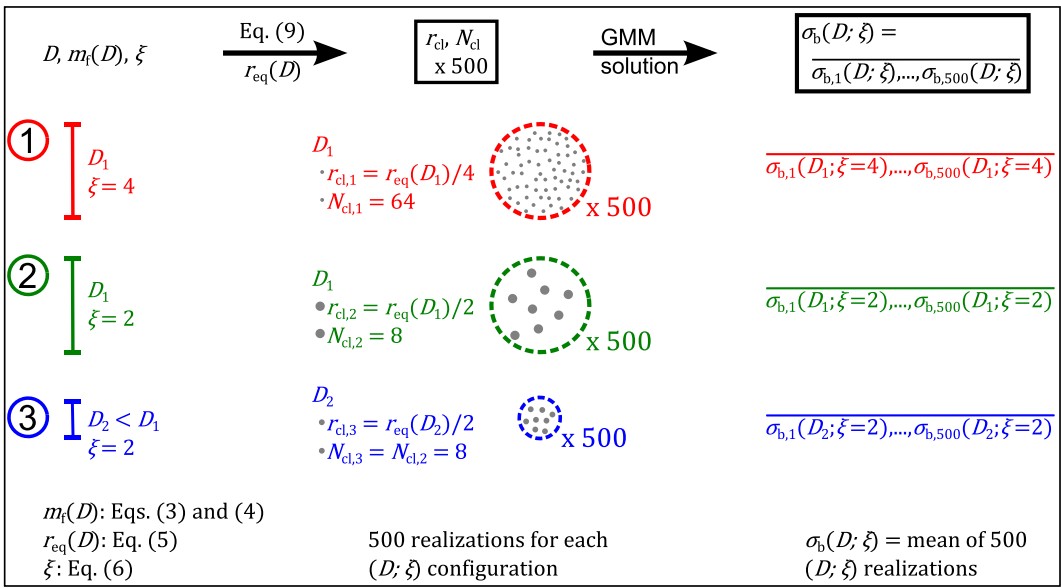

**Figure 5.**

tions. Figure 3 showed power-law exponents of $b \lesssim 1$, leading to $q \lesssim 1$ according to Eq. (8). Nonetheless, $\xi(D)$ with $q \gg 1$ are also included for completeness. Section 4.2 indicates that these $\xi(D)$ relationships contribute only a small fraction to the total range of modeled snowfall triple-frequency radar signatures and do not affect the drawn conclusions.

### 3.3 Snowflake backscatter cross sections

5 Microwave backscatter by a snowflake is modeled at X-, Ku-, Ka-, and W-band frequencies of 10, 14, 35, and 94 GHz, respectively. Here, the (radar) backscatter cross section $\sigma_b$ is calculated for mass- and SAV-equivalent collections of non-overlapping ice spheres with the generalized multiparticle Mie (GMM) solution (Xu, 1995; Xu and Å. S. Gustafson, 2001). Calculated $\sigma_b$ values correspond to the differential scattering cross sections at backscatter multiplied by $4\pi$ (see Bohren and Huffman (1983) for a discussion on commonly applied conventions for expressing backscatter by a particle). The modeling approach is outlined 10 in Fig. 5 and described in this section.

A snowflake defined by the diameter $D$, the mass $m_f(D)$, and the normalized surface-area-to-volume ratio $\xi$ is represented by a collection of ice spheres where the radius $r_{cl}$ and the number $N_{cl}$ of the constituent ice spheres are specified by $m_f$, or equivalently by $r_{eq}$ through Eq. (5), and by $\xi$:

$$r_{cl}(r_{eq}, \xi) = \frac{r_{eq}}{\xi} \,,$$

15 $$N_{cl}(\xi) = \xi^3 \,. \tag{9}$$

The snowflake diameter $D$ specifies the (spherical) bounding volume $V_f$ of the collection of ice spheres according to Eq. (4). Limitations of this representation and implications for the modeling results presented in Sect. 4 are discussed in Appendix A.

Equation (9) preserves snowflake mass and normalized surface-area-to-volume ratio given by $m_\mathrm{f} = \frac{4\pi}{3}\rho_\mathrm{ice}r_\mathrm{eq}^3 = \frac{4\pi}{3}\rho_\mathrm{ice}N_\mathrm{cl}r_\mathrm{cl}^3$ and $\xi = \mathsf{SAV}_\mathrm{f}/\mathsf{SAV}_\mathrm{s} = (\frac{3}{r_\mathrm{cl}})/(\frac{3}{r_\mathrm{eq}})$, respectively. This approach derives the parameterization of the constituent ice spheres from effective microstructural properties of the modeled snowflake in contrast to other methods where mass and shape of the constituent ice crystals were parameterized in detail and the microstructure of the modeled snowflake was then derived by aggregation of the ice crystals (e.g., Westbrook et al., 2004; Nowell et al., 2013; Leinonen and Moisseev, 2015).

The MASC observations presented in Sect. 2.2 showed nearly uniform distributions of snowflake orientation angles and therefore suggest randomly oriented snowflakes for the analyzed snowfall data. To account for random snowflake orientation in the applied modeling approach and also include a variety of 3D microstructures derived for the same values of $D$, $m_\mathrm{f}(D)$, and $\xi$, 500 realizations of randomly distributed non-overlapping ice spheres inside $V_\mathrm{f}$ are used to model each configuration of $D$ and $\xi$, or equivalently each configuration of $r_\mathrm{cl}$ and $N_\mathrm{cl}$. The snowflake backscatter cross section $\sigma_\mathrm{b}(D;\xi)$ is determined as the mean of all backscatter cross sections $\sigma_{\mathrm{b},1}(D;\xi),...,\sigma_{\mathrm{b},500}(D;\xi)$ that are calculated by the GMM solution for the 500 individual realizations. Here, the refractive index of all constituent ice spheres is given by the complex refractive index $n_{\mathrm{ice},\lambda}$ of pure ice calculated according to Mätzler and Wegmüller (1987), leading to refractive indices of $n_{\mathrm{ice},\lambda} = 1.8 + 2.3 \cdot 10^{-4}i$, $1.8 + 3.2 \cdot 10^{-4}i$, $1.8 + 8.2 \cdot 10^{-4}i$, and $1.8 + 2.4 \cdot 10^{-3}i$ (rounded to two significant figures) at 10, 14, 35, and 94 GHz, respectively.

Sets of 500 realizations were chosen for averaging because mean values of $\sigma_\mathrm{b}(D;\xi)$ stabilize to within relative differences of less than 0.1 once $10^1$ to $10^2$ collections of randomly distributed ice spheres are included (see Fig. S2 in the Supplement for details). These uncertainties in $\sigma_\mathrm{b}(D;\xi)$ are small compared to the impact of $\xi$ on modeled $\sigma_\mathrm{b}(D;\xi)$, characterized by relative differences of up to a factor of $10^2$ in Sect. 4.1. The presented methodology then quantifies the impact of normalized SAV on the calculated backscatter cross sections without including effects due to the spatial distribution or clustering of the $N_\mathrm{cl}$ ice spheres inside the bounding volume $V_\mathrm{f}$.

To analyze the impact of snowflake surface-area-to-volume ratio on modeled backscatter cross sections for a given snowflake diameter $D$, $\sigma_\mathrm{b}(D;\xi)$ are calculated for seven values of $N_\mathrm{cl} = 1, 4, 8, 16, 27, 64$, and 125, corresponding to normalized snowflake surface-area-to-volume ratios of $\xi = 1.0, 1.6, 2.0, 2.5, 3.0, 4.0$, and 5.0, respectively. Backscatter cross sections for intermediate values of $N_\mathrm{cl} = \xi^3$ are determined from linear interpolations of the seven calculated $\sigma_\mathrm{b}(D;\xi)$ values. The parameter $\xi$, describing the snowflake microstructure, and the number $N_\mathrm{cl}$, specifying the corresponding collections of randomly distributed ice spheres inside the snowflake bounding volume, are used interchangeably throughout this study according to Eq. (9).

For comparison, the analysis also includes mass-equivalent soft (mixed ice–air) oblate spheroids and snowflakes modeled according to the self-similar Rayleigh–Gans approximation (SSRGA; Hogan and Westbrook, 2014; Hogan et al., 2017). Backscatter cross sections of randomly oriented soft spheroids with major axis length $D$ are calculated with the T-matrix method (Waterman, 1971), using the implementation of Mishchenko and Travis (1998) within the PyTMatrix software package of Leinonen (2014). Aspect ratios of $\alpha = 1, 0.6$, and 0.2 are considered, representing soft spheres with $\alpha = 1$, spheroids that are characterized by typical average values of $\alpha = 0.6$ found in snowflake observations (e.g., Korolev and Isaac, 2003; Gergely and Garrett, 2016) and used for the interpretation of snow- and ice-cloud radar measurements (Matrosov et al., 2005; Hogan et al., 2012), and spheroids described by extreme values of observed snowflake aspect ratios of $\alpha = 0.2$. Effective re-

fractive indices of the soft spheroids are determined by applying the Maxwell–Garnett mixing rule (Maxwell Garnett, 1904) for volume mixtures of ice inclusions in air, given by the mass $m_\mathrm{f}(D)$ and the volume $\alpha V_\mathrm{f}$ of the spheroidal snowflakes, and for the complex refractive index $n_{\mathrm{ice},\lambda}$ of pure ice. The SSRGA has been derived to approximate backscatter cross sections for detailed 3D shape models of aggregate snowflakes based on a statistical description of mean snowflake microstructure and deviation

from the mean microstructure. Calculated $\sigma_\mathrm{b}$ values with the SSRGA represent ensemble averages for $10^1$ different realizations of the snowflake 3D microstructure with the same snowflake diameter $D$, for 50 random orientations of each snowflake 3D shape model, and for then illuminating each of the reoriented 3D shape models along its three orthogonal directions. Here, the SSRGA is applied to snowflake masses derived by Eqs. (3) and (4) and for complex refractive indices $n_{\mathrm{ice},\lambda}$ of pure ice, using the parameterizations listed by Hogan et al. (2017) for synthetic aggregate snowflakes that were generated according to

Westbrook et al. (2004), abbreviated as 'W04' throughout the text, and according to Nowell et al. (2013), abbreviated as 'N13'.

### 3.4 Snowfall triple-frequency radar signatures

In this study, snowfall triple-frequency radar signatures are defined by the two dual-wavelength ratios of modeled snowfall radar reflectivity factors at (i) Ka and W band and at (ii) either X and Ka band or Ku and Ka band, where X, Ku, Ka, and W band refer to frequencies of 10, 14, 35, and 94 GHz, respectively. The selected frequencies are within ±1 GHz of X-, Ku-, Ka-,

and W-band frequencies commonly used for the analysis of snowfall triple-frequency radar signatures (Leinonen et al., 2012; Kulie et al., 2014; Kneifel et al., 2015, 2016; Yin et al., 2017).

To derive snowfall triple-frequency radar signatures at X, Ka, and W band and at Ku, Ka, and W band, snowflake (radar) backscatter cross sections $\sigma_\mathrm{b}$ modeled according to Sect. 3.3 are first integrated for exponential snowflake size distributions $N(D)$ expressed through Eq. (1), yielding the corresponding snowfall (equivalent) radar reflectivity factors $Z_e$ (e.g.,

Matrosov, 2007; Liu, 2008):

$$Z_e = \frac{\lambda^4}{\pi^5} \left| \frac{n_{\mathrm{w},\lambda}^2 + 2}{n_{\mathrm{w},\lambda}^2 - 1} \right|^2 \int\limits_0^{D_\mathrm{max}} \sigma_\mathrm{b}(D;\xi) N(D) dD \; , \tag{10}$$

where $n_{\mathrm{w},\lambda}$ denotes the complex refractive index of liquid water at wavelengths of $\lambda = 30.0$, 21.4, 8.6 and 3.2 mm for the analyzed frequencies of 10, 14, 35, and 94 GHz, respectively. Here, $n_{\mathrm{w},\lambda}$ is determined for pure water at a temperature of $0\,^\circ\mathrm{C}$ following Meissner and Wentz (2004).

Snowfall triple-frequency radar signatures are then given by dual-wavelength ratios (DWRs, Kneifel et al., 2011) of

$$\mathrm{DWR}\ \lambda_1/\lambda_2 = 10 \cdot \log_{10}\left( \frac{Z_{e,\lambda_1}}{Z_{e,\lambda_2}} \right) \tag{11}$$

$$= \mathrm{dB}Z_{e,\lambda_1} - \mathrm{dB}Z_{e,\lambda_2} \; , \tag{12}$$

where $\lambda_1/\lambda_2$ indicate the pairs of analyzed radar frequency bands of X/Ka, Ku/Ka, and Ka/W.

Radar reflectivity factors $Z_e$ are calculated by Eq. (10) for snowflake diameters of $D \leq D_\mathrm{max} = 23.6$ mm, or for mass-

equivalent ice sphere radii of $r_\mathrm{eq} \leq 2.1$ mm according to Eqs. (3)–(5). This snowflake diameter range covers more than $99.99\,\%$

of all MASC observations presented in Sect. 2.2. Furthermore, snowflake size distributions $N(D)$ given by Eq. (1) with exponential slope parameters of $0.3 \leq \Lambda \leq 5.0$ mm$^{-1}$ are included in the analysis. This range of $\Lambda$ covers all $N(D)$ determined from the MASC observations that were presented in Sect. 2.2, corresponds to size distributions derived from snowflake observational data that were collected with different measurement methods (e.g., Brandes et al., 2007; Tiira et al., 2016), and is similar to $\Lambda$ ranges used in prior studies that have modeled snowfall triple-frequency radar signatures (e.g., Kneifel et al., 2011).

In Sect. 4.2, snowfall triple-frequency radar signatures are also modeled for size distributions limited to snowflake diameters of $D \leq 10.0$ mm and $D \leq 5.0$ mm. The corresponding triple-frequency radar signatures are derived by applying the presented modeling approach for modified snowflake maximum diameters of $D_{\mathrm{max}} = 10.0$ mm and $D_{\mathrm{max}} = 5.0$ mm.

## 4 Modeling results and discussion

### 4.1 Snowflake backscatter cross sections

Figure 6 shows snowflake backscatter cross sections $\sigma_{\mathrm{b}}$ modeled according to Sect. 3.3 at 35 and 94 GHz and for snowflake diameters of $D \leq 14.4$ mm, corresponding to mass-equivalent ice sphere radii of $r_{\mathrm{eq}} \leq 1.5$ mm. The total range of $\sigma_{\mathrm{b}}$ for all diameters of $D \leq 23.6$ mm, for all considered snowflake models, and for frequencies of 10 and 14 GHz is included in Fig. S3 in the Supplement.

For soft spheres, Figs. 6 and S3 show strong resonances in calculated $\sigma_{\mathrm{b}}$ typical for applying Mie scattering theory to large particles (Mie, 1908; Bohren and Huffman, 1983). The higher the frequency, and thus the larger the effective size of a spherical particle with diameter $D$ relative to the wavelength, the more oscillations are observed within the total diameter range. Oscillations in $\sigma_{\mathrm{b}}$ are heavily dampened for spheroids due to orientation averaging of $\sigma_{\mathrm{b}}$ and for SSRGA results due to averaging over an ensemble of many different realizations of non-spherical snowflake shape models. Collections of randomly distributed ice spheres inside the (spherical) snowflake bounding volume also lead to a much weaker oscillation pattern in $\sigma_{\mathrm{b}}$ than soft spheres of diameter $D$ because the refractive index $n_{\mathrm{ice},\lambda}$ of pure ice generally differs significantly from the effective refractive indices of soft spheres determined with the Maxwell–Garnett mixing rule (real and imaginary parts of soft-sphere effective refractive indices are smaller and thus closer to one and zero, respectively) and because the ice spheres are characterized by a radius of $r_{\mathrm{cl}} \ll D/2$ and therefore by a much smaller effective size relative to the wavelength (see Sect. 3.3).

In Fig. 6, calculated backscatter cross sections $\sigma_{\mathrm{b}}(D;\xi)$ for collections of $1 \leq N_{\mathrm{cl}} = \xi^3 \leq 125$ randomly distributed ice spheres inside the snowflake bounding volume cover a maximum range of over 2 orders of magnitude for $r_{\mathrm{eq}} \approx 0.85$ mm or $D \approx 6.3$ mm at 35 GHz and for $r_{\mathrm{eq}} \approx 0.44$ mm or $D \approx 2.4$ mm at 94 GHz. Outside the Mie resonance regions, $\sigma_{\mathrm{b}}(D;\xi)$ decrease with increasing normalized surface-area-to-volume ratio $\xi$. This trend is consistent with results of Honeyager et al. (2014) who found smaller backscatter cross sections for greater snowflake surface complexity when modeling microwave backscatter for their snowflake 3D shape models with the discrete dipole approximation.

A comparison of the $\sigma_{\mathrm{b}}$ curves in Figs. 6 and S3 shows that differences in $\sigma_{\mathrm{b}}$ associated with the choice of snowflake model generally increase with increasing snowflake diameter and microwave frequency. In Fig. 6, $\sigma_{\mathrm{b}}$ curves can be distinguished

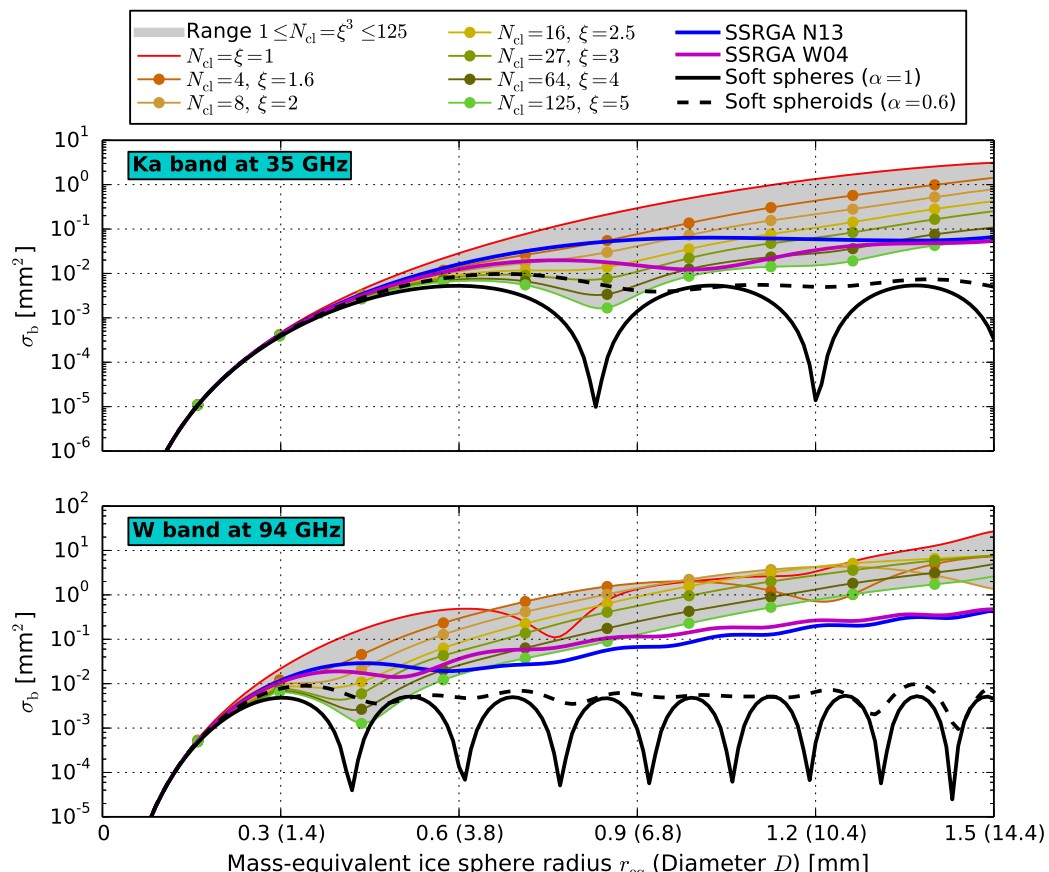

**Figure 6.**

visibly from each other for $r_{eq} > 0.3$ mm or $D > 1.4$ mm at 35 GHz while $\sigma_b$ curves already split for $r_{eq} \approx 0.2$ mm or $D \approx 0.6$ mm at 94 GHz, for example. SSRGA results for the N13 and W04 snowflake parameterizations are similar to each other and fall within the indicated range of $\sigma_b(D; \xi)$ for collections of $1 \leq N_{cl} = \xi^3 \leq 125$ randomly distributed ice spheres for small snowflake diameters and low microwave frequencies. For large snowflake diameters and high frequencies, however,

5  backscatter cross sections $\sigma_b$ calculated by the SSRGA are up to 1 order of magnitude smaller than the minimum $\sigma_b(D; \xi)$. Compared to soft spheres, $\sigma_b$ values calculated by the SSRGA are up to 4 orders of magnitude higher in Fig. 6.

The N13 and W04 snowflake parameterizations according to the SSRGA used in this study were originally derived for snowflake 3D shape models with diameters $D \lesssim 10$ mm by Hogan et al. (2017). Nonetheless, these SSRGA parameterizations are applied to snowflake diameters up to $D_{max} = 23.6$ mm in the presented analysis to allow a direct comparison with modeled

10  backscatter by collections of randomly distributed ice spheres and by soft spheres and spheroids (this extension of the SSRGA validity range is briefly discussed below).

Diameters of $D = 10.0$ mm and $D = 5.0$ mm, indicated in Fig. S3 by vertical dashed lines, are used as maximum diameters $D_{\max}$ for the analysis of truncated snowflake size distributions in Sect. 4.2. Combined with the analysis of modeled snowfall triple-frequency radar signatures for $D_{\max} = 23.6$ mm, the results for snowflake size distributions truncated at $D_{\max} = 10.0$ mm and at $D_{\max} = 5.0$ mm then characterize the impact of large snowflakes with $D > 10.0$ mm and with $D > 5.0$ mm on modeled snowfall triple-frequency radar signatures.

Notably, snowfall triple-frequency radar signatures modeled according to the SSRGA for N13 and W04 snowflake parameterizations and snowflake size distributions truncated at $D_{\max} = 10.0$ mm in Sect. 4.2 show similar characteristic differences with respect to triple-frequency radar signatures modeled for collections of randomly distributed ice spheres and for soft spheres and spheroids as the differences found for snowflake size distributions spanning the total analyzed range of diameters up to $D_{\max} = 23.6$ mm. Therefore, application of the two SSRGA snowflake parameterizations beyond the size range they were originally derived for by Hogan et al. (2017) is not expected to significantly affect the corresponding analysis results and conclusions in this study.

## 4.2 Snowfall triple-frequency radar signatures

An overview of the snowfall radar reflectivity factors $Z_e$ derived from the modeled snowflake backscatter cross sections in Sect. 4.1 is included in Fig. S4 in the Supplement but not discussed in this study. Snowfall triple-frequency radar signatures are shown in Fig. 7. For all considered snowflake models, using DWR Ku/Ka to quantify triple-frequency radar signatures in combination with DWR Ka/W leads to compressed triple-frequency curves by $\Delta$DWR $\lesssim 3$ dB compared to using DWR X/Ka. But the general shape of each curve and characteristic differences among the shapes of all curves are not affected by the choice of defining triple-frequency radar signatures with respect to either DWR X/Ka or DWR Ku/Ka.

Triple-frequency curves for soft spheres and spheroids with aspect ratios of $\alpha = 1$, 0.6, and 0.2 in Fig. 7 are characterized by strictly increasing DWRs with decreasing exponential slope parameters $\Lambda$ of the snowflake size distribution. For a given value of $\Lambda$, DWRs determined for the three aspect ratios are generally within 3 dB from each other.

Modeled triple-frequency radar signatures for the N13 and W04 snowflake parameterizations according to the SSRGA roughly follow the shape of the curves determined for soft spheres and spheroids for high values of $\Lambda$, but show a maximum in DWR Ka/W near $\Lambda \approx 0.5$ mm$^{-1}$. A further decrease of $0.5 \geq \Lambda \geq 0.3$ mm$^{-1}$ then leads to a decrease in DWR Ka/W by less than 1 dB, resulting in triple-frequency curves roughly shaped like a comma sign. Based on synthetic aggregate snowflakes generated according to W04, Stein et al. (2015) related the maximum in DWR Ka/W to the fractal geometry of the modeled aggregate snowflakes.

For collections of randomly distributed ice spheres inside the (spherical) snowflake bounding volume, triple-frequency curves in Fig. 7 derived for low normalized surface-area-to-volume ratios of $\xi \approx 1$ show only a slow increase in DWR X/Ka or DWR Ku/Ka with decreasing $\Lambda$ and therefore occupy a region below the triple-frequency curves determined for soft spheres and spheroids and for the N13 and W04 snowflake parameterizations. Increasing values of $\xi$ lead to curves that follow the shapes of the triple-frequency curves derived for soft spheres and spheroids with $\alpha = 1$ and $\alpha = 0.6$ for narrow snowflake size distributions characterized by high values of $\Lambda$. However, triple-frequency curves derived for collections of randomly

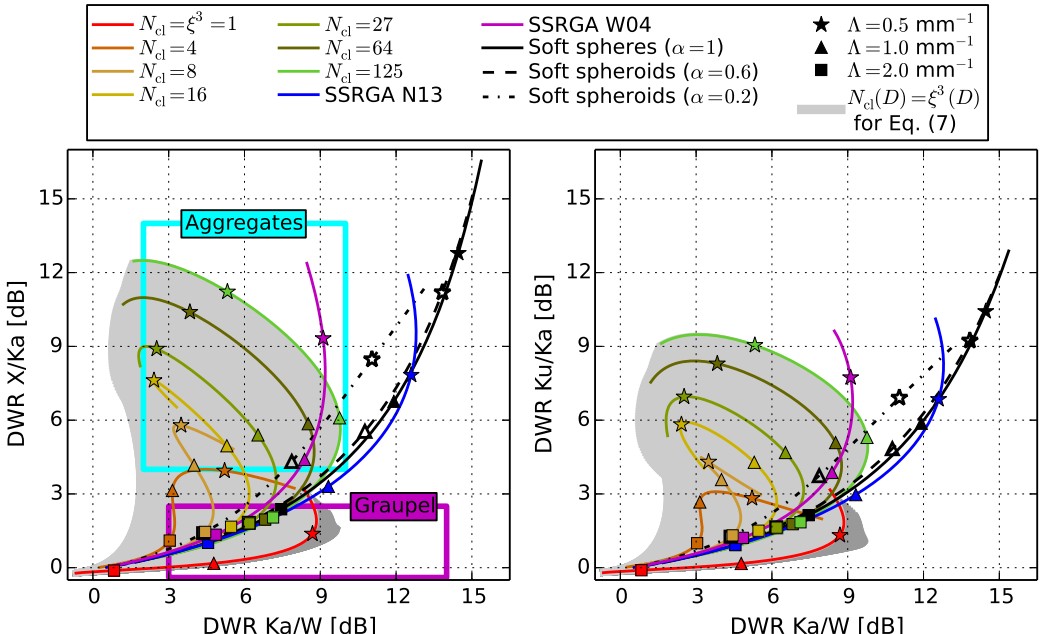

**Figure 7.**

distributed ice spheres generally reach a strong maximum in DWR Ka/W at an intermediate value of $\Lambda$ and then sharply bend back toward lower DWR Ka/W with a further decrease in $\Lambda$. This behavior leads to hook-shaped triple-frequency curves. The strength of the 'hooking' increases with increasing $\xi$, quantified through the difference between maximum DWR Ka/W and the value of DWR Ka/W corresponding to the minimum slope parameter of $\Lambda = 0.3$ mm$^{-1}$. Additionally, higher values of $\xi$ result

in triple-frequency curves that roughly follow the shape of spheroidal curves up to higher values of DWR X/Ka or DWR Ku/Ka before hooking toward lower DWR Ka/W (see also Fig. 8 for triple-frequency curves determined for $\xi = 6$).

The hook shape of triple-frequency curves derived for intermediate and high normalized surface-area-to-volume ratios $\xi$ in Fig. 7 is similar to the general shape of snowfall triple-frequency curves that were previously modeled by Kneifel et al. (2011) and Leinonen et al. (2012) based on non-spheroidal snowflake 3D shape models. Neither soft spheres and spheroids nor

the N13 and W04 snowflake parameterizations according to the SSRGA yield triple-frequency curves showing such a strong maximum in DWR Ka/W at intermediate values of $\Lambda$.

Modeling snowfall triple-frequency radar signatures for collections of randomly distributed ice spheres inside the snowflake bounding volume also leads to a much wider range of triple-frequency radar signatures in Fig. 7 than the region between the triple-frequency curves derived for soft spheres and spheroids or for the N13 and W04 snowflake parameterizations according

to the SSRGA. Modeled triple-frequency curves for $1 \leq \xi \leq 5$ cover a range of up to about 10 dB in DWR X/Ka, 8 dB in DWR Ku/Ka, and 7 dB in DWR Ka/W (see also Fig. S5). In contrast, soft spheres and spheroids or the N13 and W04 snowflake parameterizations according to the SSRGA show DWR ranges of generally about 3 dB and less.

The total range of triple-frequency radar signatures modeled for collections of randomly distributed ice spheres in Fig. 7 covers a large part of all observed triple-frequency signatures in snowfall radar reflectivity measurements by Kulie et al. (2014), Kneifel et al. (2015), and Yin et al. (2017). This modeled range also includes many of the triple-frequency radar signatures that Stein et al. (2015) observed in their radar reflectivity measurements at 3, 35, and 94 GHz and modeled based on synthetic aggregate snowflakes generated according to W04. In the present study, the overlap between modeled W04 triple-frequency curve and the total range of triple-frequency radar signatures modeled for collections of randomly distributed ice spheres increases for small $\Lambda$, and thus for broad snowflake size distributions characterized by larger snowflakes, when higher normalized surface-area-to-volume ratios of $\xi > 5$ are also included in the modeling approach (see Fig. 8 for the effect of including $\xi = 6$).

Modeled triple-frequency radar signatures in Fig. 7 for intermediate and high values of $\xi$ combined with small exponential slope parameters $\Lambda$ of the snowflake size distribution correspond to triple-frequency radar signatures that were related to the presence of large aggregate snowflakes by Kneifel et al. (2015). The region of triple-frequency radar signatures that they related to snowfall characterized by rimed snowflakes, denoted as graupel in Fig. 7, contains triple-frequency curves modeled for low normalized surface-area-to-volume ratios of $\xi \approx 1$ in this study. High values of $\xi$ indicate high complexity of the snowflake microstructure (Sect. 3.2), as expected for aggregate snowflakes. Furthermore, broad snowflake size distributions characterized by small $\Lambda$ in Eq. (1) contain a higher amount of large snowflakes, consistent with the observation of large aggregates for triple-frequency radar signatures that correspond to small $\Lambda$. Extensive snowflake riming, on the other hand, is associated with a coarsening or rounding of the snowflake microstructure due to the accretion of supercooled water droplets. This reduction in the complexity of the snowflake microstructure for strongly rimed snowflakes is reflected in the applied modeling approach by low normalized surface-area-to-volume ratios, leading to relatively flat triple-frequency curves with consistently low dual-wavelength ratios DWR X/Ka and DWR Ku/Ka for $\xi \approx 1$.

In contrast, snowfall triple-frequency radar signatures that were modeled by Leinonen and Szyrmer (2015) based on detailed 3D shape models of rimed snowflakes extend to higher values of DWR X/Ka and DWR Ku/Ka and roughly span the region between the W04 and N13 triple-frequency curves shown in Fig. 7 for small exponential slope parameters $\Lambda$, depending on the amount of riming assigned to the snowflake 3D shape models. Nonetheless, Fig. S6 indicates that truncated size distributions of the rimed snowflake 3D shape models analyzed by Leinonen and Szyrmer (2015), i.e., snowflake size distributions excluding large snowflakes, again lead to flat triple-frequency curves characterized by consistently low DWR X/Ka and DWR Ku/Ka, in line with the snowfall triple-frequency radar signatures related to snowflake riming by Kneifel et al. (2015) and modeled for low normalized snowflake surface-area-to-volume ratios of $\xi \approx 1$ in this study (see also the discussion below of how truncated snowflake size distributions and different parameterizations of snowflake mass affect modeled snowfall triple-frequency radar signatures).

Triple-frequency curves determined for soft spheres and spheroids and for the N13 and W04 snowflake parameterizations according to the SSRGA cover a much smaller region of the indicated range of observed snowfall triple-frequency radar signatures in Fig. 7 than the triple-frequency radar signatures modeled for collections of randomly distributed ice spheres inside the snowflake bounding volume and do not explain the distinct regions related to the presence of large aggregates and rimed snowflakes that were observed by Kneifel et al. (2015). Notably, even if various combinations of snowflake gamma size

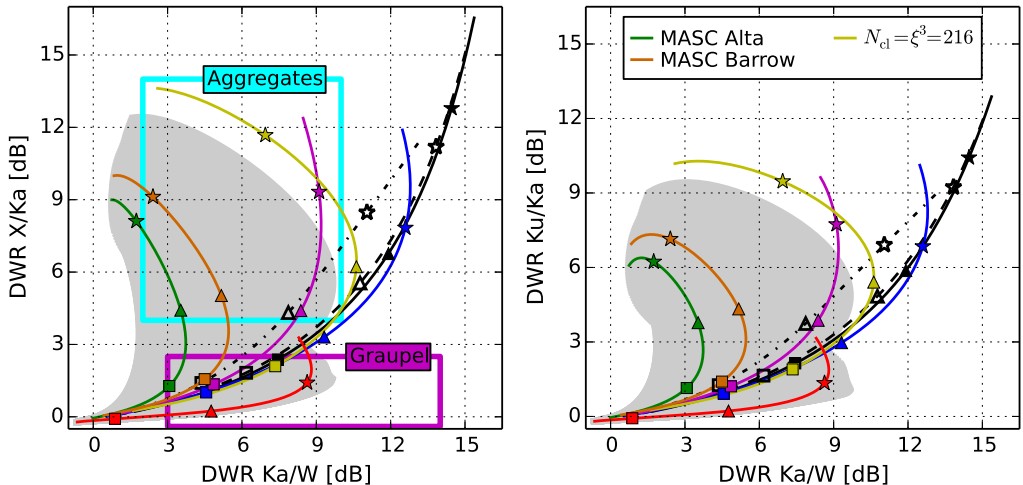

**Figure 8.**

distributions, mass–diameter relationships, aspect ratios, and distributions of preferentially horizontally oriented snowflakes are used to model snowfall triple-frequency radar signatures for soft spheroids, the range of modeled triple-frequency radar signatures does not show significantly better agreement with the observed range of snowfall triple-frequency signatures in radar reflectivity measurements (Leinonen et al., 2012; Kneifel et al., 2015).

Comparing radar reflectivity measurements and in situ snowflake observations, Kneifel et al. (2015) also found that a clear distinction between different snow types was not feasible for combinations of low DWR Ka/W and low DWR X/Ka. Here, this ambiguity can be explained by the similarity of all triple-frequency curves in Fig. 7 for high exponential slope parameters $\Lambda$, and thus for narrow snowflake size distributions according to Eq. (1). Modeled triple-frequency radar signatures for narrow snowflake size distributions are dominated by small snowflakes; and for small snowflakes, the differences in the modeled

snowflake backscatter cross sections shown in Figs. 6 and S3 are not significant enough to cause a clear separation of the modeled triple-frequency curves in Fig. 7 at high $\Lambda$. For larger snowflakes, larger differences among modeled backscatter cross sections are found in Figs. 6 and S3. As broader snowflake size distributions characterized by lower values of $\Lambda$ contain a higher amount of large snowflakes, the modeled triple-frequency curves in Fig. 7 are more easily distinguished at small $\Lambda$.

Modeled snowfall triple-frequency radar signatures based on the MASC measurements of snowflake complexity $\chi$ presented

in Sect. 2.2 are shown in Fig. 8. The two $\xi(D)$ relationships derived by inserting Eq. (2) into Eq. (8), with fitted exponents of $q = b = 0.75$ for the MASC data recorded at Alta and $q = b = 0.54$ for the Barrow data, still lead to hook-shaped triple-frequency curves with a maximum in DWR Ka/W at intermediate values of $\Lambda$. However, the maximum value of DWR Ka/W is smaller and the hook shape is therefore less pronounced than for triple-frequency curves derived for constant normalized snowflake surface-area-to-volume ratios of $\xi \gtrsim 3$ in Fig. 7.

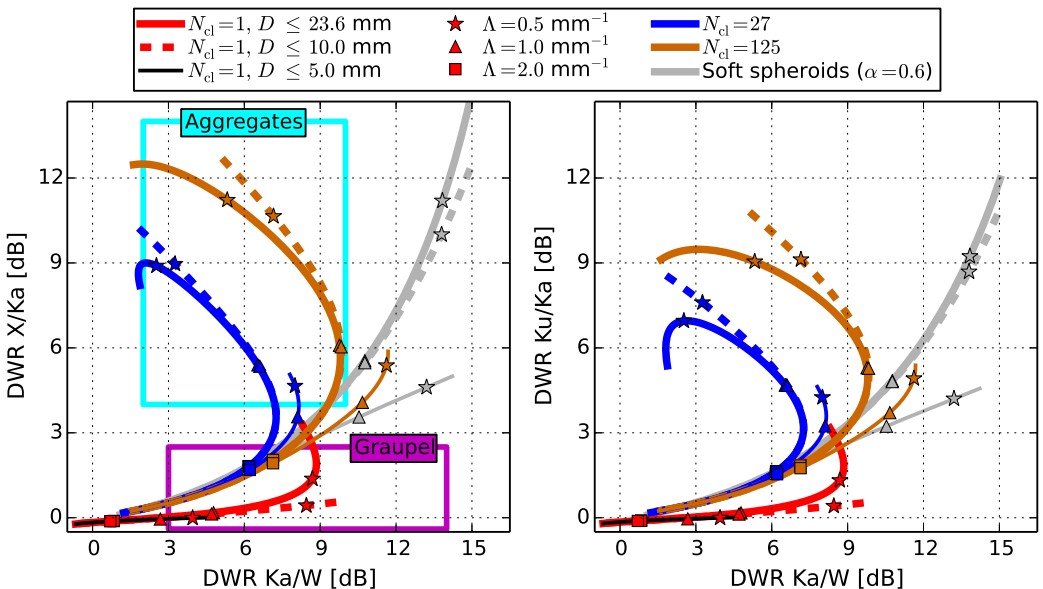

**Figure 9.**

Thus far, all snowfall radar signatures have been determined for exponential snowflake size distributions with snowflake diameters of $D \leq D_{\mathrm{max}} = 23.6$ mm. To investigate the effect of truncating snowflake size distributions already at smaller maximum diameters, snowfall triple-frequency radar signatures were also modeled for exponential snowflake size distributions limited to $D \leq D_{\mathrm{max}} = 10.0$ mm and $D \leq D_{\mathrm{max}} = 5.0$ mm. The modeling results are presented in Figs. S7 and S8 in the
Supplement and summarized in Fig. 9.

In general, truncation at smaller $D_{\mathrm{max}}$ leads to an 'un-hooking' or flattening of the derived triple-frequency curves. For $D \leq$ 10.0 mm, modeled snowfall triple-frequency radar signatures in Fig. 9 follow the corresponding triple-frequency curves derived for $D \leq 23.6$ mm down to snowflake size distributions characterized by exponential slope parameters of $\Lambda \approx 1.0$ mm$^{-1}$ before splitting off (toward higher values of DWR Ka/W for $N_{\mathrm{cl}} = \xi^3 = 27, \; 125$ and toward lower DWR X/Ka and DWR Ku/Ka
for $N_{\mathrm{cl}} = \xi = 1$). Triple-frequency curves derived for $D \leq 5.0$ mm already start to deviate visibly from the two corresponding curves determined for $D \leq 23.6$ mm and for $D \leq 10.0$ mm at higher values of $\Lambda \approx 2.0$ mm$^{-1}$. Additionally, truncating snowflake size distributions at $D_{\mathrm{max}} = 5.0$ mm leads to a smaller total range of modeled DWR X/Ka and DWR Ku/Ka, consistent with DWR modeling results presented by Kneifel et al. (2011) for truncated snowflake size distributions of various snowflake 3D shape models. For low normalized surface-area-to-volume ratios, indicated by $N_{\mathrm{cl}} = \xi = 1$ in Fig. 9, trunca-
tion at $D_{\mathrm{max}} = 5.0$ mm also leads to a smaller range of modeled DWR Ka/W. The comparison of the triple-frequency curves in Fig. 9 shows the strong impact of the maximum diameter $D_{\mathrm{max}}$ of the snowflake size distribution on modeled snowfall triple-frequency radar signatures.

For snowflake size distributions limited to diameters of $D \leq D_{\max} = 10.0$ mm, modeled snowfall triple-frequency radar signatures based on the MASC measurements of snowflake complexity $\chi$ at Alta and at Barrow are included in Fig. S9. Compared to Fig. 8, truncation at $D_{\max} = 10.0$ mm leads to an increase in modeled DWRs of up to about 3 dB. These differences are caused by the strong influence of $D_{\max}$ on the value of $\chi(D_{\max}) = \chi_{\max}$ calculated with Eq. (2), i.e., $\chi_{\max}$
of the two $\chi(D)$ relationships illustrated in Fig. 3 decreases for snowflake size distributions truncated at smaller $D_{\max}$, which translates into higher normalized snowflake surface-area-to-volume ratios $\xi(\chi)$ for $D \leq D_{\max} = 10.0$ mm following Eq. (8). A reliable determination of $D_{\max}$ is therefore also important for modeling snowfall triple-frequency radar signatures based on snowflake complexity measurements.

     Combining the hook shape of triple-frequency curves derived for high normalized surface-area-to-volume ratios in Figs. 7
and 8 with the flattening of triple-frequency curves due to the truncation of snowflake size distributions at smaller maximum diameters as illustrated in Fig. 9, modeled triple-frequency radar signatures for snowfall characterized by high snowflake surface-area-to-volume ratios and small snowflake diameters can resemble snowfall triple-frequency radar signatures modeled for soft spheroids. This explains why some non-spheroidal snowflake shape models may lead to similarly high values of modeled DWR Ka/W > 10 dB as soft spheroids, e.g., for the aggregates of needle-shaped ice crystals analyzed by
Leinonen et al. (2012). According to Fig. 9, values of DWR Ka/W > 10 dB are expected for snowfall characterized by normalized snowflake surface-area-to-volume ratios of $\xi \approx 5$ and exponential snowflake size distributions limited to snowflake diameters of $D \leq D_{\max} = 5.0$ mm with exponential slope parameters of $\Lambda \lesssim 1.0$ mm$^{-1}$. Higher values of $\xi > 5$ already lead to DWR Ka/W > 10 dB for less restrictive snowflake size distributions with respect to $D_{\max}$ and $\Lambda$.

     All presented results have been determined for only one parameterization of snowflake mass $m_{\mathrm{f}}(D)$ according to Sect. 3.1.
Previous studies have shown, however, that the uncertainty in modeled snowfall radar reflectivity factors $Z_e$ due to the parameterization of $m_{\mathrm{f}}(D)$ is significant. Hammonds et al. (2014) found uncertainties in $Z_e$ related to $m_{\mathrm{f}}(D)$ on the order of 4 dB at X, Ku, Ka, and W band, for example. To evaluate the impact of the parameterization of snowflake mass on the modeled snowfall triple-frequency radar signatures in this study, DWRs for collections of $N_{\mathrm{cl}} = 1,\ 27,\ 125$ randomly distributed ice spheres inside the snowflake bounding volume (corresponding to normalized surface-area-to-volume ratios of $\xi = 1,\ 3,\ 5$)
were also derived after uniformly increasing and decreasing the density values $\rho_{\mathrm{f}}(D)$ obtained from the H04 density–diameter relationship, and thus the snowflake masses $m_{\mathrm{f}}(D)$ given by Eqs. (3) and (4), by 25 % and by 50 %. Derived triple-frequency curves for the modified $\rho_{\mathrm{f}}(D)$ are shown in Figs. S10 and S11, and the impact of the parameterization of snowflake mass on modeled $Z_e$ and DWRs is summarized in Fig. 10.

     The analyzed $\rho_{\mathrm{f}}(D)$ range leads to a corresponding range in modeled $Z_e$ of $\Delta\mathrm{dB}Z_e > 3.5$ dB and a range in derived DWRs
of $\Delta\mathrm{DWR} < 3.0$ dB in Fig. 10. Generally, differences of $\Delta\mathrm{dB}Z_e \gtrsim 6$ dB and of $\Delta\mathrm{DWR} \lesssim 1$ dB are found, except for snowfall characterized by $\xi = 1$, indicative of heavily rimed graupel snow according to Fig. 7, and snowflake size distributions with exponential slope parameters of $\Lambda \lesssim 2$ mm$^{-1}$. Similar trends are also noted for snowflake size distributions limited to $D \leq D_{\max} = 10.0$ mm and $D \leq D_{\max} = 5.0$ mm (not shown). Here, an increase (decrease) in $\rho_{\mathrm{f}}(D)$ for snowfall characterized by $\xi = 1$ additionally yields consistently higher (lower) DWR Ka/W for all $\Lambda$ and thus an increase (decrease) in the
modeled DWR Ka/W range (see Fig. S11 for truncation at $D_{\max} = 10.0$ mm; and extreme differences $\Delta\mathrm{DWR}$ for $\xi = 1$ are

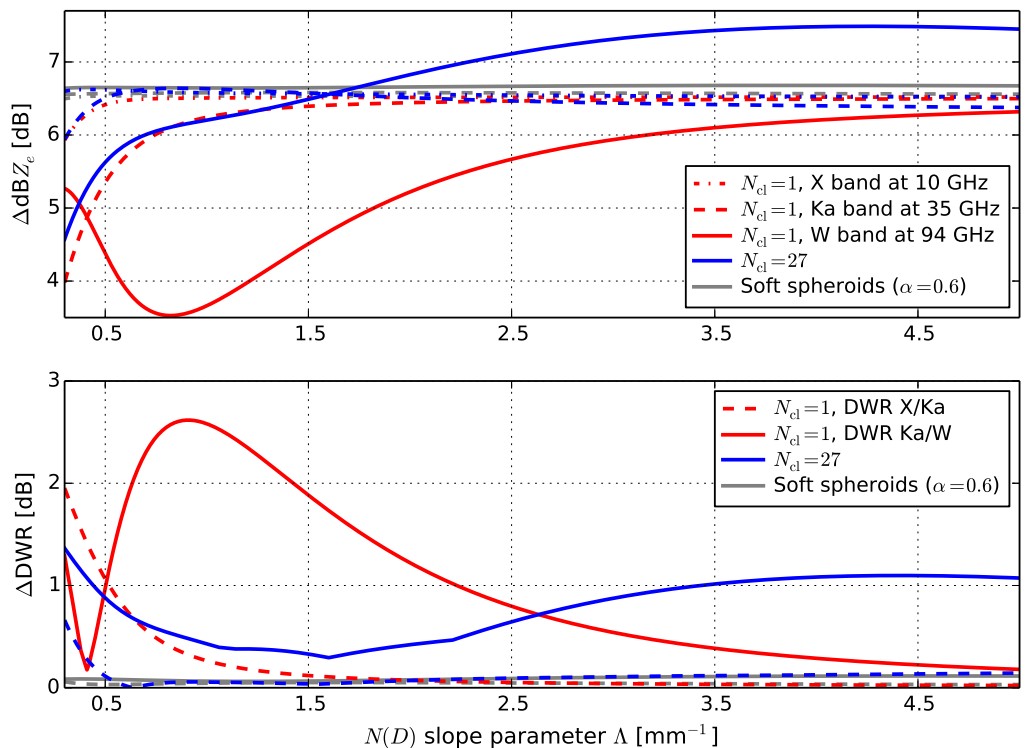

**Figure 10.**

illustrated in Fig. S12 by comparing triple-frequency radar signatures determined for the H04 snowflake density–diameter relationship with triple-frequency radar signatures determined for a snowflake mass–diameter relationship which was derived by Locatelli and Hobbs (1974) specifically to describe lump graupel). These results demonstrate that modeled DWRs are less sensitive to uncertainties associated with the parameterization of snowflake mass than modeled $Z_e$ at a single wavelength but

5   can still be affected significantly by these uncertainties, especially at low normalized surface-area-to-volume ratios.

Nonetheless, even high differences of $\Delta$DWR $> 1$ dB associated with changes in $\rho_f(D)$ and $m_f(D)$ of $\pm 50\,\%$ are generally much smaller than the differences $\Delta$DWR of up to about 10 dB in DWR X/Ka, 8 dB in DWR Ku/Ka, and 7 dB in DWR Ka/W associated with the range of normalized surface-area-to-volume ratios of $1 \leq \xi \leq 5$ (compare Fig. 10 with Fig. S5). The presented analysis then highlights the importance of snowflake surface-area-to-volume ratio for a detailed interpretation of ob-

10   served and modeled snowfall triple-frequency radar signatures.

## 5 Conclusions

In this study, snowflake (radar) backscatter cross sections were modeled at X-, Ku-, Ka-, and W-band radar frequencies of 10, 14, 35, and 94 GHz based on representing individual snowflakes by collections of randomly distributed ice spheres. The size and number of the constituent ice spheres are defined by the snowflake mass derived from the snowflake maximum dimension or diameter $D$ and by the snowflake surface-area-to-volume ratio (SAV); the bounding volume of each collection of ice spheres is given by a sphere of diameter $D$. SAV was quantified through the normalized ratio $\xi$ of snowflake SAV to the SAV of a single mass-equivalent ice sphere for a range of $1 \leq \xi \leq 5$.

Snowfall triple-frequency radar signatures were then determined from dual-wavelength ratios (DWRs) of the snowfall equivalent radar reflectivity factors $Z_e$ that were calculated using the modeled snowflake backscatter cross sections. Based on near-surface snowflake observations collected by high-resolution multi-view imaging at Alta (UT, USA) and at Barrow (AK, USA), $Z_e$ and DWRs were calculated for exponential snowflake size distributions with snowflake diameters of $D \leq D_{\max} = 23.6$ mm and exponential slope parameters of $0.3 \leq \Lambda \leq 5.0$ mm$^{-1}$.

The analysis focused on the impact of snowflake SAV on modeled snowfall triple-frequency radar signatures. Additionally, snowflake complexity values obtained from the snowflake images and averaged over one winter season were used as an indicator of snowflake SAV to derive snowfall triple-frequency radar signatures at Alta and at Barrow. Finally, the effect of truncating snowflake size distributions at $D_{\max} = 10.0$ mm and at $D_{\max} = 5.0$ mm on modeled triple-frequency radar signatures was investigated, and the impact of the parameterization of snowflake mass on modeled DWRs was evaluated by uniformly increasing and decreasing all snowflake densities, and thus all snowflake masses, by up to 50 %.

Important findings are summarized by the following bullet points:

– Average snowflake complexity increases with increasing snowflake size.

– Modeled snowflake backscatter cross sections generally decrease with increasing snowflake surface-area-to-volume ratio (SAV).

– Modeled snowfall triple-frequency radar signatures cover a wide range of snowfall triple-frequency signatures previously determined from radar reflectivity measurements.

– Snowflake SAV and truncated snowflake size distributions offer a physical interpretation of snowfall triple-frequency radar signatures that is consistent with previously observed differences in snowfall triple-frequency radar signatures related to the presence of large aggregate snowflakes and rimed snowflakes and that may explain why some snowfall triple-frequency radar signatures apparently point to a spheroidal snowflake shape.

– While modeled $Z_e$ show high sensitivity to the parameterization of snowflake mass, with typical differences of $\Delta \mathrm{dB} Z_e \gtrsim$ 6 dB for the analyzed snowflake density range, derived DWRs are less sensitive, with corresponding differences of $\Delta \mathrm{DWR} \lesssim 1$ dB except for low SAV.

– The analyzed impact of the parameterization of snowflake mass on modeled snowfall triple-frequency radar signatures is generally much smaller than the analyzed impact of snowflake SAV.

Overall, the results indicate a strong influence of snowflake SAV on modeled snowfall radar signatures that may be exploited in the interpretation of snowfall triple-frequency radar measurements, e.g., to distinguish snow types characterized by different snowflake SAV. For a detailed analysis of snowfall triple-frequency radar signatures based on snowflake SAV, however, a more comprehensive quantification of snowflake SAV will be needed. This should include characteristic differences and similarities in snowflake SAV among various snow types and reveal potential relationships between snowflake SAV and other microstructural parameters important for the interpretation of snowfall radar signatures like snowflake size and mass.

Accordingly, current and future databases of microwave scattering properties determined for detailed snowflake 3D shape models would benefit from incorporating snowflake surface area as additional microstructural parameter (besides snowflake size and mass). Common features and differences in modeled scattering properties could then be related not only to visually distinct snow types (and snowflake size and mass) but also to snowflake surface-area-to-volume ratio, providing a quantitative description of the snowflake microstructure across all snow types and thereby helping to further constrain snowflake shape for snowfall remote sensing.

Based on a more comprehensive quantification of snowflake surface-area-to-volume ratio that reflects characteristic differences among snow types, the outlined approach for relating normalized snowflake surface-area-to-volume ratio $\xi$ to snowflake complexity $\chi$ obtained from snowflake images could be applied to derive $\xi(D)$ relationships for a variety of snowfall conditions. Snowfall triple-frequency radar signatures could then be modeled from these $\xi(D)$ relationships and compared to triple-frequency radar reflectivity measurements. Such comparisons would show whether $\xi(D)$ relationships derived from snowflake imgaging data can adequately describe snowflake surface-area-to-volume ratio for the interpretation of snowfall triple-frequency radar signatures and may therefore lead to a parameterization of snowflake shape by $\xi(D)$ relationships similar to the parameterization of snowflake mass by density–diameter or mass–diameter relationships commonly used in snowfall remote sensing.

## 6 Data availability

Modeled snowflake backscatter cross sections and dual-wavelength ratios of snowfall equivalent radar reflectivity factors are included in the Supplement. Additional data may be obtained by contacting the corresponding author.

## Appendix A:  Representation of snowflakes by collections of randomly distributed ice spheres

In this study, snowflakes defined by the maximum dimension or diameter $D$, the mass $m_f(D)$, and the normalized surface-area-to-volume ratio $\xi$ are represented by collections of randomly distributed ice spheres where the radius $r_{cl}$ and the number $N_{cl}$ of the constituent ice spheres are specified by Eq. (9) and the diameter of the (spherical) bounding volume $V_f$ of each ice sphere

collection is given by $D$. The Appendix discusses limitations of this representation and implications for the modeled radar signatures.

To generate collections of non-overlapping ice spheres inside $V_f$ according to Eq. (9), $\xi^3 = N_{cl}$ has to be an integer and the snowflake mass $m_f(D)$ has to be sufficiently low. Backscatter cross sections $\sigma_b(D;\xi)$ were calculated for collections of

$N_{cl} = 1, 4, 8, 16, 27, 64,$ and 125 ice spheres, corresponding to normalized surface-area-to-volume ratios of $\xi = 1.0, 1.6, 2.0, 2.5, 3.0, 4.0,$ and 5.0, respectively (see Sect. 3.3). Backscatter cross sections for all intermediate values, integers and non-integers $\xi^3$, were determined from linear interpolations. These interpolated $\sigma_b(D;\xi)$ were used in Sect. 4.2 to outline the total range of modeled snowfall triple-frequency radar signatures for $1 \leq \xi \leq 5$ and to derive triple-frequency curves for the two sets of MASC observations at Alta and at Barrow. Most of the discussion in Sect. 4, however, focused on ice sphere

collections characterized by the seven $N_{cl}$ or corresponding $\xi$ values with calculated $\sigma_b(D;\xi)$. Uncertainties associated with the interpolation of $\sigma_b(D;\xi)$ for $1 \leq \xi \leq 5$ should therefore play only a minor role in the presented analysis.

To determine radar reflectivity factors $Z_e$ with Eq. (10), $\sigma_b(D;\xi)$ for collections of multiple ice spheres were calculated only for snowflake diameters of $D > 0.55$ mm, corresponding to (single) mass-equivalent ice sphere radii of $r_{eq} > 0.16$ mm. For smaller snowflakes, Eqs. (3) and (4) lead to high snowflake masses that could not be reached consistently by randomly

placing non-overlapping ice spheres given by Eq. (9) inside the snowflake bounding volume $V_f$. Here, $\sigma_b$ was calculated only for a single mass-equivalent ice sphere specified by $\xi = 1$, and the value of $\sigma_b(D;\xi = 1)$ was then assigned to all ice sphere collections, leading to $\sigma_b(D; 1 \leq \xi \leq 5) = \sigma_b(D; \xi = 1)$ for $D \leq 0.55$ mm or $r_{eq} \leq 0.16$ mm. This simplification has no significant impact on modeled snowfall triple-frequency radar signatures in Sect. 4.2 because radar reflectivity factors determined with Eq. (10) are only affected weakly by the backscatter cross sections of small snowflakes. Even when snowflake

diameters of $D \leq 0.55$ mm are ignored completely, modeled $Z_e$ decrease and DWRs increase by less than about 0.3 dB for snowflake size distributions with exponential slope parameters of $\Lambda \leq 2.0$ mm$^{-1}$. Slightly higher changes in modeled $Z_e$ and DWRs are noted for snowflake size distributions characterized by higher values of $\Lambda$, with a maximum decrease of 1.7 dB in modeled $Z_e$ at 94 GHz and a maximum increase of 0.8 dB in DWR Ka/W found for an extreme slope parameter of $\Lambda = 5.0$ mm$^{-1}$. These differences are generally much smaller than the impact of normalized SAV on modeled $Z_e$ and DWRs

discussed in Sect. 4.2.

At 10 and 14 GHz, all $\sigma_b(D; 1 < \xi \leq 5)$ for $0.55 < D \leq 1.4$ mm or $0.16 < r_{eq} \leq 0.3$ mm were additionally replaced by $\sigma_b(D;\xi = 1)$ to obtain smooth spline interpolants of $\sigma_b(D;\xi)$ across the entire range of $D$ (see Sect. 4.1). The effect of these modifications on modeled snowfall triple-frequency radar signatures in Sect. 4.2 is again small, with associated differences in modeled $Z_e$ and in DWR X/Ka and DWR Ku/Ka of less than about 0.1 dB for $\Lambda \leq 2.0$ mm$^{-1}$ and slightly increasing

differences for higher $\Lambda$ up to a maximum of 0.7 dB at $\Lambda = 5.0$ mm$^{-1}$.

As $N_{cl}$ non-overlapping ice spheres were placed randomly inside the spherical bounding volume $V_f$ specified by the snowflake diameter $D$ (see Sect. 3.3), the maximum dimension or diameter $D_{cl}$ of each generated ice sphere collection is always smaller than $D$ (also note that $D_{cl} \neq 2r_{cl}$; $r_{cl}$ quantifies the size of each individual ice sphere within the collection, $D_{cl}$ indicates the size of the entire collection). Figure S13 illustrates the relation between snowflake diameter $D$ and the mean

diameter $\overline{D}_{cl}$ of 500 generated collections of randomly distributed ice spheres inside $V_f$. High values of $N_{cl} = \xi^3$, i.e., col-

lections of many small ice spheres, lead to small relative differences between $D$ and $\overline{D}_{\mathrm{cl}}$ of less than $5\,\%$. For collections of $N_{\mathrm{cl}} = 4$ (and thus fewer but larger) ice spheres, $\overline{D}_{\mathrm{cl}}$ is up to about $25\,\%$ smaller than the snowflake diameter. Nonetheless, the calculated backscatter cross sections $\sigma_{\mathrm{b}}$ for the ice sphere collections show only a weak correlation with the diameter $D_{\mathrm{cl}}$ (see examples in Fig. S14), and Fig. S15 illustrates the weak influence of the differences between $D$ and $D_{\mathrm{cl}}$ on the modeled snowfall triple-frequency radar signatures. Here, dual-wavelength ratios DWR X/Ka, DWR Ku/Ka, and DWR Ka/W generally change by less than 1 dB when the mean diameter $\overline{D}_{\mathrm{cl}}$ of the generated ice sphere collections is used instead of the snowflake diameter $D$ to determine the corresponding snowfall radar reflectivity factors $Z_e$ with Eq. (10). These differences are again small compared to differences in DWRs associated with the range of normalized surface-area-to-volume ratios of $1 \le \xi \le 5$.

*Competing interests.* T. J. Garrett is a member of the editorial board of the journal and has a financial interest in Particle Flux Analytics which sells the MASC.

*Acknowledgements.* M. Gergely's work was supported through NASA grant NNX14AP78G and by the German Research Foundation (DFG) through DFG research fellowship GE 2658/1-1; S. J. Cooper acknowledges support from the National Science Foundation (NSF) under grant 1531930; T. J. Garrett was supported through NSF grant 1303965 and US Department of Energy grant DE-SC0016282. The authors thank H. Löwe at the WSL Institute for Snow and Avalanche Research SLF and two anonymous referees for their comments that helped improve this study.

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

**Figure captions**

List of figure captions:

**Figure 1.** (left) MASC single-view images of two snowflakes: (top) aggregate snowflake and (bottom) heavily rimed graupel snow. (right) Illustration of the corresponding projection images of perimeter $P$ (highlighted white regions) and area-equivalent circles of circumference $C$ (outlined in red), leading to complexity values $\chi = \frac{P}{C}$ of (top) $\chi = 2.1$, (bottom) $\chi = 1.2$. Derived snowflake diameters $D$ and orientation angles $\theta$ are indicated by solid and dotted magenta lines, respectively: (top) $D = 5.7$ mm, $\theta = 16°$; (bottom) $D = 2.3$ mm, $\theta = 31°$.

**Figure 2.** Snowflake (frequency) size distributions $N(D)$ and relative distributions of snowflake complexity $\chi$ and orientation $\theta$ for $4.4 \cdot 10^5$ snowflakes sampled by MASC at Alta and Barrow. Dashed lines represent minimum and maximum slope parameters $\Lambda_{min}$ and $\Lambda_{max}$ of exponential snowflake size distributions $N(D)$ and exponential complexity distributions $N(\chi)$ fitted to 47 snowstorms at Alta and to 7 snowstorms at Barrow. The number of recorded extreme values outside the plotted range is 33 for $D$ and 43 for $\chi$. Mean orientation angles at Alta and at Barrow are $\overline{\theta} = 40°$ and $\overline{\theta} = 45°$, respectively. Numerical values of $\Lambda_{min}$, $\Lambda_{max}$, and mean $\overline{\Lambda}$ are given in the text.

**Figure 3.** Logarithmic 2D histogram for all MASC data of snowflake diameter $D$ and complexity $\chi$ presented in Fig. 2, with bin sizes of $\Delta D = 0.1$ mm and $\Delta\chi = 0.01$. Mean complexity values per size bin are indicated by $\overline{\chi}/\Delta D$ for snowflake data recorded at Alta and at Barrow separately. Snowflake complexity–diameter relationships $\chi(D)$ for the data sets collected at Alta and at Barrow are determined by the non-linear least squares method for fitting Eq. (2) to the values of $\overline{\chi}/\Delta D$ and characterized by the power-law exponent $b$.

**Figure 4.** Synthetically generated $\xi(D)$ relationships for deriving normalized snowflake surface-area-to-volume ratio $\xi$ from snowflake diameter $D$ by Eq. (7) with $D \leq D_{max} = 23.6$ mm. Shown $\xi(D)$ curves reflect the total range of $\xi(D)$ relationships used for modeling snowfall triple-frequency radar signatures in Sect. 4.2.

**Figure 5.** Sketch of the modeling approach described in Sect. 3.3, with three examples highlighted in red, green, and blue. The impact of normalized snowflake surface-area-to-volume ratios of $1 \leq \xi \leq 5$ on modeled snowflake backscatter cross sections $\sigma_b$ is investigated by applying the generalized multiparticle Mie (GMM) solution to collections of randomly distributed ice spheres characterized by the radius $r_{cl}$ and the number $N_{cl}$ of the constituent ice spheres and by the snowflake diameter $D$ indicating the spherical bounding volume of the ice sphere collections.

**Figure 6.** Modeled snowflake backscatter cross sections $\sigma_b$ at 35 and 94 GHz for (i) collections of $1 \leq N_{cl} \leq 125$ randomly distributed ice spheres inside a spherical bounding volume of diameter $D$, corresponding to normalized surface-area-to-volume ratios of $1 \leq \xi \leq 5$, for (ii) the self-similar Rayleigh–Gans approximation (SSRGA) applied to N13 and to W04 snowflake 3D shape models, and for (iii) soft spheres and oblate spheroids with aspect ratios of $\alpha = 1$ and $\alpha = 0.6$, respectively. Results for (single) mass-equivalent ice spheres given by $N_{cl} = 1$, for snowflakes modeled according to the SSRGA, and for soft spheres and spheroids were calculated at a resolution of $\Delta r_{eq} = 0.01$ mm. For collections of $N_{cl} = 4, 8, 16, 27, 64,$ and 125 ice spheres, dots mark values of $\sigma_b(D;\xi)$ that were calculated at a resolution of $\Delta r_{eq} \approx 0.14$ mm following Sect. 3.3, and lines indicate spline interpolations of the calculated $\sigma_b(D;\xi)$. Modeled $\sigma_b$ for the full range of considered snowflake diameters $D \leq 23.6$ mm, for soft spheroids characterized by extreme aspect ratios of $\alpha = 0.2$, and for microwave frequencies of 10 and 14 GHz are shown in Fig. S3 in the Supplement.

**Figure 7.** Modeled snowfall triple-frequency radar signatures given by dual-wavelength ratios of DWR Ka/W and either DWR X/Ka or DWR Ku/Ka. DWRs are determined according to Sect. 3.4 for exponential size distributions characterized by snowflake diameters of $D \leq 23.6$ mm and exponential slope parameters of $0.3 \leq \Lambda \leq 5.0$ mm$^{-1}$. Snowflakes are represented by (i) collections of randomly distributed ice spheres inside the spherical snowflake bounding volume, by (ii) the N13 and W04 snowflake parameterizations according to the self-similar

Rayleigh–Gans approximation (SSRGA), and by (iii) soft spheres and oblate spheroids. The gray area indicates the plume of all triple-frequency curves derived for collections of randomly distributed ice spheres that are described by synthetically generated $\xi(D)$ relationships expressed through Eq. (7) and summarized in Fig. 4. Darker shade of gray marks the region of DWR combinations derived for high power-law exponents of $q > 2.5$ in Eq. (7). Colored rectangles are adapted from Kneifel et al. (2015) and roughly outline regions associated with the presence of large aggregate snowflakes (cyan) and rimed snowflakes (graupel; magenta) that were inferred by relating snowfall triple-frequency radar reflectivity measurements at X, Ka, and W band to coincident in situ snowflake observations. Corresponding triple-frequency radar signatures for snowflake size distributions limited to $D \leq 10.0$ mm and to $D \leq 5.0$ mm are shown in Figs. S7 and S8, respectively.

**Figure 8.** Modeled snowfall triple-frequency radar signatures for exponential size distributions with snowflake diameters of $D \leq 23.6$ mm. The notation follows Fig. 7 with additional DWRs determined for ice sphere collections with a normalized surface-area-to-volume ratio of $\xi = 6$ and by applying Eq. (8) to the MASC measurement results from Alta and from Barrow presented in Fig. 3. Corresponding triple-frequency radar signatures for exponential size distributions limited to $D \leq 10.0$ mm are shown in Fig. S9.

**Figure 9.** Impact of snowflake maximum diameter $D_{\mathrm{max}} = 23.6$, 10.0, 5.0 mm on modeled snowfall triple-frequency radar signatures for exponential snowflake size distributions with exponential slope parameters of $0.3 \leq \Lambda \leq 5.0$ mm$^{-1}$. Modeling results for collections of $N_{\mathrm{cl}} = 1$, 27, 125 randomly distributed ice spheres inside the spherical snowflake bounding volume correspond to normalized surface-area-to-volume ratios of $\xi = 1$, 3, 5, respectively.

**Figure 10.** Impact of the parameterization of snowflake mass on modeled snowfall radar reflectivity factors $Z_e$ and dual-wavelength ratios (DWRs) for exponential size distributions $N(D)$ with snowflake diameters of $D \leq 23.6$ mm and exponential slope parameters of $0.3 \leq \Lambda \leq 5.0$ mm$^{-1}$. Shown $\Delta \mathrm{dB}Z_e$ and $\Delta$DWR curves indicate the maximum difference in derived dB$Z_e$ values and DWRs that is associated with uniformly increasing and decreasing all snowflake densities $\rho_{\mathrm{f}}(D)$ obtained from the H04 density–diameter relationship, and thus all snowflake masses $m_{\mathrm{f}}(D)$ determined from Eqs. (3) and (4), by 25 % and by 50 %. Modeling results for dB$Z_e$ at 14 GHz and for DWR Ku/Ka are similar to shown dB$Z_e$ at 10 GHz and DWR X/Ka, respectively. Collections of $N_{\mathrm{cl}} = 125$ ice spheres, corresponding to a normalized surface-area-to-volume ratio of $\xi = 5$, lead to similar $\Delta \mathrm{dB}Z_e$ and $\Delta$DWR as the included ice sphere collections with $N_{\mathrm{cl}} = 27$ or $\xi = 3$.