# Peer review of "Using snowflake surface-area-to-volume ratio to model and interpret snowfall triple-frequency radar signatures"

_Atmospheric Chemistry and Physics, 2017_

## Referee Comment (RC1) · Anonymous Referee #1 · 12 Jun 2017

Review of "Using snowflake surface area-to-volume ratio to model and interpret snowfall triple-frequency radar signatures" by Mathias Gergely, Steven J Cooper, and Timothy J Garrett.

Summary:

The authors present relationships between particle effective diameter, mass, and complexity, based on several days of snowstorm observations from two different locations. The 2D observation of complexity is then applied to construct 3D realisations of snowflakes, based on individual spherical elements, to subsequently use in scattering calculations. Compared to other snowflake scattering models, the authors find

that their 3D realisations – taking into account the variety of snowflake complexity – better cover the observed range of of radar scattering values given by the radar dual wavelength ratio.

General review:

This is a very well-written and well-presented study, covering a new aspect of snowflake modelling from an insightful perspective. The use of triple-wavelength observations for snowfall retrievals is very promising and it is important to have original studies such as this to consider the various snowflake characteristics that influence the radar signatures. Given the quality and the novelty, this paper should be considered for publication subject to minor revisions.

Minor comments:

1. SAV versus complexity

The section from p.11, line 25 to p.13, line 3, is difficult to follow. It reads as if ksi (SAVf/SAVs) is a completely new characteristic of snowflake structure, but in many ways it is simply a 3D consideration of the complexity, chi. Instead of perimeter divided by area (1D versus 2D), SAV considers area divided by volume (2D versus 3D). This discussion culminates in equation 11, where indeed ksi is shown to be uniquely related to chi. This set of paragraphs would be better placed in section 3.2, perhaps after p.10, line 15. Alternatively, the set could be entirely removed, as it does not seem to add much to the discussion. In particular, it is not clear where in section 4.2 the result is shown that variation of the exponent q has some effect on radar scattering.

2. SSRGA snowflake parameterization validation

The authors appear to suggest that the W04 and N13 SSRGA snowflake parameterizations are not supported by observations in terms of their DWR curves. However, the W04 model was shown to compare very well against triple-frequency observations of stratiform ice clouds in southern England (Stein et al. 2015). Stein et al. (2015)

[Figure]

furthermore relate the "maximum in DWR Ka/W" (p.17, line 1) to the fractal nature of aggregate snowflakes. Similarly, "the indicated range of observed snowfall triple-frequency radar signatures" on p.18, line 4, does not consider the southern England study. The authors should therefore also rephrase "this characteristic behaviour with a strong maximum of DWR Ka/W at intermediate values of Lambda", as it may not be universal.

Stein, T. H. M., C. D. Westbrook, and J. C. Nicol (2015), Fractal geometry of aggregate snowflakes revealed by triple-wavelength radar measurements, Geophys. Res. Lett., 42, 176–183, doi:10.1002/2014GL062170.

3. Riming

The authors' consideration of how riming affects snowflake scattering behaviour and the resulting triple-wavelength DWR curves appears to conflict with the simulations of Leinonen and Szyrmer (2015). In that study, the DWR curves look like a selection of curves spanning the range between the W04 and N13 curves in the current Figure 7. This contrasts starkly with the authors' expectation that riming leads to a curve similar to ksi=1, with low DWR Ku/Ka. The authors should comment on the findings of Leinonen and Szyrmer (2015) in relation to their discussion of the effects of riming on snowflake characteristics. (Also p.17, line 33 onwards).

Leinonen, J., and W. Szyrmer (2015), Radar signatures of snowflake riming: A modeling study, Earth and Space Science, 2, 346–358, doi:10.1002/2015EA000102.

4. Practical application

The first avenue for future research (page 22, line 4-8) would indeed be "interesting", but it seems rather impractical. Having unique ksi(D) relationships for each individual snowfall event would not be useful for NWP model development or even microphysical modelling studies, as it would simply be too much effort to implement. It would also seem rather impractical for operational snowfall rate retrievals. The authors should

provide slightly more detail on how their research could be applied in practice.

Selected other comments:

Figure 2. There 's a lot going on in these figures. The numbers in the bottom right should be removed and instead directly quoted in the caption. The average orientation should be removed as well (grey box in bottom left panel) and quoted in the caption.

p.6, line 10: Why is it necessary to combine the data into a single data set for complexity, but not for N(D)?

p.6 line 14 and line 20: The distribution in figure 3 seems skewed. The median should be a better measure of typical complexity rather than the mean.

p.8 line 10: Is this modification done randomly?

p.9, line 5: What does Delta mean here?

Figure 6. Again rather busy. The lines at 5mm and 10mm with boxed saying D=5mm and D=10mm are unnecessary, even though they are referred to in the text.

p.18, line 27: "the maximum of DWR is already found at lower values of DWR" – this is confusing, possibly a typo and Lambda is meant?

p.18, line 33: The "un-hooking" makes the curve behave more like the W04 and N13 curves in Figure 8. Similarly, the truncation makes the N=125 curve in Figure 9 behave more like the W04 and N13 curves as well. Not sure what to make of this.
* * *

---

## Referee Comment (RC2) · Anonymous Referee #2 · 12 Jun 2017

The manuscript presents a very interesting study on how additional descriptors of ice particles can be used to better constrain a connection between scattering and physical snowflake properties. The manuscript is well written and with exception of a few minor problems is easy to understand. Because I would like to see the authors response to several of my comments, I would like to suggest to publish the paper if the authors address those concerns adequately.

Major comments:

1. The authors are modeling snowflakes as collection of solid ice spheres with a prescribed mass, D and SAV. How realistic this assumption is? Leinonen and Moisseev

(2015) have argued that using spheroids (or spheres) of solid ice in place of the crystals leads to the formation of much denser aggregates. Or in the other words is it possible to match mass, D and SAV of a realistic snowflake using a set of spheres?

Leinonen, J., and D. Moisseev (2015), What do triple-frequency radar signatures reveal about aggregate snowflakes?, J. Geophys. Res. Atmos., 120, doi:10.1002/2014JD022072.

2. While modelling scattering from soft spheroids, the authors have used the assumptions that ice particles are randomly oriented. This assumption is supported by the presented observations of orientation angles as shown in Fig. 2. However, this assumption contradicts dual-polarization and multi-frequency radar observations, see work of Matrosov et al for example. For example, differential reflectivity values, Zdr, characteristic of aggregates lie in the range from 0 to 1 dB. This range can be reproduced by soft spheroids with aspect ratio of 0.6 and a preferential horizontal orientation. If the random orientation is assumed the expected Zdr value would 0 dB. A possible explanation of the discrepancy is the difference in an optical and microwave definition of particle shape. Imagine that an ice particle consists of a horizontally aligned spheroid and an attached dendritic crystal, such that the crystal orientation angle is different from 0. If most of the spheroid mass is much larger than that of the dendrite than for radar scattering calculations the particle can be assumed to be spheroidal and horizontally aligned. The shadow image of the particle would be different from the spheroid, and the orientation angle of this complex particle is different from 0. At the moment, we don't know what is the best assumption of a particle shape and what is the relation between optical and microwave particle properties of ice particles. Therefore, we should use models that covers a larger range of possible backscattering properties. I suggest that instead of random orientation the authors would use a spheroid with a preferential horizontal orientation.

3. The authors state that they use observations from 47 snowstorms observed in Utah and 7 storms in Barrow, which resulted in $4.3 \cdot 10^5$ and $10^4$ snowflake observations.

The number of snowflakes sounds to be too small. I would expect that $4.3 \cdot 10^5$ would be a good number of snowflakes recorded during a single snowstorm. Of course, this number depends on the instrument sampling volume and how often observations are made. Both of which are not discussed in the paper. Could you please include more information on how the measurements are made, how PSD are computed, how often images are taken, etc.

Minor comments: 1. There is a lot of discussion about the snowflake complexity, while the main focus of the paper on SAV. It is a little bit confusing? Could you consider them together or explain how you compute SAV from observations?

2. In the paper, the snowflake complexity parameter is used to describe ice particle properties. On page 6. the authors mention that for particles with D>=3 mm the complexity parameter is larger than 1, which corresponds to aggregates. What about heavily rimed aggregates? Would the complexity parameter and SAV be different from 1? It is not directly related to this study, but I have seen other studies where this parameter is used as an indicator of riming. I would be interesting to know, whether this parameter can be used as a riming indicator for all types of particles, regardless of their initial complexity.

---

## Author Comment (AC1) · 23 Aug 2017

Dear Editor,

We would like to thank the two anonymous Referees for their comments, which helped us clarify the presented analysis.

Based on the Referees' comments, additional details have been provided in the revised manuscript, especially on the MASC measurement method and analysis (in Sect. 2) and on limitations of the applied method for representing snowflakes by collections of ice spheres and implications for the presented modeling results (new Appendix A). Furthermore, the introduction of the normalized snowflake surface-area-to-volume ratio xi has been moved to a new section (new Sect. 3.2).

We enclose point-by-point responses to the comments from the Referees including page and line numbers that indicate changes in the revised manuscript. Additions and modifications to the previous manuscript in response to the comments from the Referees are also highlighted by blue text in the marked-up .pdf version of the revised manuscript. Changes due to the reorganization of the text and modifications that were made throughout the text for clarification and to maintain consistency are not highlighted.

With best regards,

 Mathias Gergely, Steven J. Cooper, Timothy J. Garrett

**Response to Referee 1:**

Referee 1:
General review:
This is a very well-written and well-presented study, covering a new aspect of
snowflake modelling from an insightful perspective. The use of triple-wavelength
observations for snowfall retrievals is very promising and it is important to have
original studies such as this to consider the various snowflake characteristics that
influence the radar signatures. Given the quality and the novelty, this paper should
be considered for publication subject to minor revisions.

Minor comments:
1. SAV versus complexity
The section from p.11, line 25 to p.13, line 3, is difficult to follow. It reads as if
ksi (SAVf/SAVs) is a completely new characteristic of snowflake structure, but in many
ways it is simply a 3D consideration of the complexity, chi. Instead of perimeter
divided by area (1D versus 2D), SAV considers area divided by volume (2D versus 3D).
This discussion culminates in equation 11, where indeed ksi is shown to be uniquely
related to chi. This set of paragraphs would be better placed in section 3.2, perhaps
after p.10, line 15. Alternatively, the set could be entirely removed, as it does not
seem to add much to the discussion. In particular, it is not clear where in section
4.2 the result is shown that variation of the exponent q has some effect on radar
scattering.

Authors:

The discussion focuses on general features of xi(D) relationships that can be understood based on the highlighted xi(D)=constant=1,...,5 curves in the figures alone. However, the mentioned text passage describing non-constant xi(D) relationships for exponents q > 0 is important for deriving xi(D) from measurements of the snowflake 2D complexity chi(D) (and thus for the practical applicability). It also forms the basis for deriving the total range of modeled snowfall triple-frequency radar signatures (gray regions in Figs. 7 and 8). Simply using many constant xi values between 1 and 5 produces a gray area in the plots with 'fraying' toward the bottom left.

 Based on the Referee's comment, the text was rearranged and the mentioned text passage now forms a new section together with the introduction of normalized snowflake surface-area-to-volume ratio xi: Section '3.2 Snowflake surface-area-to-volume ratio' introduces xi and gives details on deriving xi(D) from averaged snowflake observational data. Additionally, the importance of non-constant xi(D) for the analysis has been clarified in the revised manuscript (p. 9 line 18ff.), and the 2 different exponents q derived for the two MASC data sets collected at Alta and at Barrow are now explicitly stated in the analysis to illustrate the effect of different q on radar scattering (p. 19 line 13f. in combination with Fig. 8).

2. SSRGA snowflake parameterization validation
The authors appear to suggest that the W04 and N13 SSRGA snowflake parameterizations
are not supported by observations in terms of their DWR curves. However, the W04 model
was shown to compare very well against triple-frequency observations of stratiform ice
clouds in southern England (Stein et al. 2015). Stein et al. (2015) furthermore relate
the "maximum in DWR Ka/W" (p.17, line 1) to the fractal nature of aggregate
snowflakes. Similarly, "the indicated range of observed snowfall triple-frequency
radar signatures" on p.18, line 4, does not consider the southern England study. The
authors should therefore also rephrase "this characteristic behaviour with a strong
maximum of DWR Ka/W at intermediate values of Lambda", as it may not be universal.
Stein, T. H. M., C. D. Westbrook, and J. C. Nicol (2015), Fractal geometry of
aggregate snowflakes revealed by triple-wavelength radar measurements, Geophys. Res.
Lett., 42, 176–183, doi:10.1002/2014GL062170.

It is not the authors' intent to suggest that snowfall triple-frequency radar signatures that are modeled based on the W04 (and N13) SSRGA snowflake parameterizations are not observed. Specifically with

regard to W04, the cyan rectangles in Figs. 7 and 8 adapted from Kneifel et al. (2015) also include the low-Lambda part of the W04 triple-frequency curve that is not included in the plotted gray region (gray region = modeled snowfall triple-frequency radar signatures using the modeling approach based on randomly distributed ice spheres with $1 <= xi(D) <= 5$ inside the snowflake bounding volume presented in this study). So, the W04 triple-frequency curve lies within the total range of observation results that are discussed in this study.

In the previous manuscript version, the study of Stein et al. (2015) was not included in the discussion of snowfall radar reflectivity observations because it uses yet another frequency to derive triple-frequency radar signatures (3 GHz instead of the analyzed 10 and 14 GHz in the manuscript), which makes a streamlined and accurate discussion difficult. Nonetheless, inclusion of their measurement results does not change any of the drawn conclusions, because many of their measurement results are also found within the range of observations given in the other discussed studies.

It is also noted that most of the W04 triple-frequency curve (and the high-Lambda part of the N13 curve) lies within the modeled total range of triple-frequency radar signatures given in the manuscript for collections of randomly distributed ice spheres with $1 <= xi(D) <= 5$ (= gray regions in Figs. 7 and 8). In Fig. 8, increasing xi (here to xi=6) further increases the overlap of the W04 curve with the modeled total range of triple-frequency signatures.

As the Referee's comment addresses the fractal nature of the aggregates, a brief statement about the fractal geometry of the aggregates has been included in the revised manuscript to explain the maximum in DWR Ka/W for W04 aggregates (p. 17 line 7ff.). Additionally, a brief discussion of how the Stein et al. (2015) observations and modeling results relate to this study is now included (p. 17 line 35ff.). Also, parts of the presented discussion were rephrased to avoid suggesting universality of the results discussed in this study, e.g., the modifier 'characteristic' is avoided when referring to triple-frequency radar signatures in the revised manuscript.

```
3. Riming
The authors' consideration of how riming affects snowflake scattering behaviour and
the resulting triple-wavelength DWR curves appears to conflict with the simulations of
Leinonen and Szyrmer (2015). In that study, the DWR curves look like a selection of
curves spanning the range between the W04 and N13 curves in the current Figure 7. This
contrasts starkly with the authors' expectation that riming leads to a curve similar
to ksi=1, with low DWR Ku/Ka. The authors should comment on the findings of Leinonen
and Szyrmer (2015) in relation to their discussion of the effects of riming on
snowflake characteristics. (Also p.17, line 33 onwards).
Leinonen, J., and W. Szyrmer (2015), Radar signatures of snowflake riming: A modeling
study, Earth and Space Science, 2, 346–358, doi:10.1002/2015EA000102.
```

Truncating snowflake size distributions of Leinonen and Szyrmer (2015)'s rimed snowflake 3D shape models already at a smaller snowflake maximum size leads to modeled snowfall triple-frequency curves characterized by consistently low DWR X/Ka and DWR Ku/Ka, similar to the triple-frequency curves modeled in this study for low normalized SAV of xi~1 and consistent with the snowfall triple-frequency radar signatures related to snowflake riming by Kneifel et al. (2015). This flattening effect of truncated snowflake size distributions on modeled triple-frequency curves for snowflake size distributions of Leinonen and Szyrmer (2015)'s rimed snowflake 3D shape models is now illustrated in Fig. S6 in the Supplement.

Following the Referee's request, a comment on the differences and similarities between this study and Leinonen and Szyrmer (2015) with respect to the discussion of the effects of snowflake riming on modeled snowfall triple-frequency radar signatures has been added to the revised manuscript (p. 18 line 18ff. and Fig. S6).

4. Practical application
The first avenue for future research (page 22, line 4-8) would indeed be
"interesting", but it seems rather impractical. Having unique ksi(D) relationships for
each individual snowfall event would not be useful for NWP model development or even
microphysical modelling studies, as it would simply be too much effort to implement.
It would also seem rather impractical for operational snowfall rate retrievals. The
authors should provide slightly more detail on how their research could be applied in
practice.

It is probably not necessary to have unique xi(D) relationships that relate normalized snowflake surface-area-to-volume ratio xi to snowflake diameter D for each individual snowfall event, but some constraints for xi(D) will have to be obtained to use these xi(D) for the analysis of snowfall radar signatures (similar to snowflake density-diameter or mass-diameter relationships, a single xi(D) relationship cannot be expected to be representative of the wide range of all snowfall conditions). Therefore, it would be desirable to better understand and quantify differences and similarities in xi among different snow types, for example. Based on a more comprehensive quantification of xi, e.g., by incorporating snowflake surface area as additional microstructural parameter into scattering databases of detailed snowflake 3D shape models, xi(D) relationships could be derived from snowflake complexity observations for a variety of snowfall conditions by the outlined approach in the manuscript and then used to model triple-frequency radar signatures. A comparison with radar reflectivity measurements should show how realistic such a description of snowflake surface-area-to-volume ratio by xi(D) relationships is for the interpretation of snowfall triple-frequency radar signatures.

The outlook in the 'Conclusions' section has been rewritten in the revised manuscript to better address the practical application of the research as discussed above (p. 24 line 8ff.).

Other comments:
Figure 2. There 's a lot going on in these figures. The numbers in the bottom right
should be removed and instead directly quoted in the caption. The average orientation
should be removed as well (grey box in bottom left panel) and quoted in the caption.

Following the Referee's request, all numerical values were removed from Fig. 2 and instead included in the caption and/or text. Additionally, Fig. 2 was modified to reflect the now consistent analysis of N(D) and N(chi) in the revised manuscript (in response to the following comment from the Referee).

p.6, line 10: Why is it necessary to combine the data into a single data set for
complexity, but not for N(D)?

Not necessary, but this choice was originally made because complexity distributions for individual snowstorms are not mentioned again in the analysis.

Based on the Referee's comment, both N(D) and N(chi) are now fitted and analyzed for individual snowstorms following the same scheme (compare p. 6 line 10f. and p. 6 line 15ff., see also updated Fig. 2).

p.6 line 14 and line 20: The distribution in figure 3 seems skewed. The median should
be a better measure of typical complexity rather than the mean.

Following the Referee's comment, the analysis was also performed based on summarizing all complexity values within each diameter bin by their median instead of their mean. The results change little, both for the fitted complexity curves in Fig. 3 and for the triple-frequency radar signatures derived for the MASC data in Fig. 8. The resulting figures based on the median instead of the mean complexity per size bin are shown below. The respective curves representing data recorded at Alta and at Barrow in Figs. 3 and 8 move closer together, and especially the complexity curve fitted to the Alta data shows worse agreement with the measurement data overall when median complexity per size bin is used to generate an alternative Fig. 3. The drawn conclusions are not affected.

The analysis in the revised manuscript is still based on the mean complexity per size bin. Nonetheless, a brief statement has been added to the text to point out that using the median complexity instead of the

mean complexity per size bin has only a minor influence on the presented analysis and does not affect the drawn conclusions (p. 6 line 23ff.).

[Figure]

No, this is done uniformly to obtain snowflake densities and thus snowflake masses that are either consistently higher or consistently lower than the densities and masses derived from the H04 density-diameter relationship.

 For clarification, 'modifying (all densities....)' was changed to 'uniformly increasing and decreasing (all densities...)' in the text for clarification (p. 8 line 18, p. 21 line 27, p. 23 line 22).

p.9, line 5: What does Delta mean here?

There is no Delta at p. 9 line 5. Probably the comment refers to Delta_sigma/sigma on p. 10 line 5. This is the relative difference between the sigma value calculated for including $10^1$ to $10^2$ realizations of the ice sphere collections for the same D and xi and the sigma value calculated for including 5000 realizations. Details are given in Fig. S2.

To avoid confusion about the definition of Delta_sigma/sigma, the expression Delta_sigma/sigma, which is not used anywhere else in the manuscript, has been deleted. For details on the relative difference that describes the stability of calculated sigma values, the reader is referred to Fig. S2 (p. 11 line 10ff.).

Figure 6. Again rather busy. The lines at 5mm and 10mm with boxed saying D=5mm and D=10mm are unnecessary, even though they are referred to in the text.

Based on the Referee's comment, the lines at D = 5 mm and at D = 10 mm and the corresponding boxes were removed from Fig. 6. They are still included in Fig. S3 in the Supplement for referencing in the text (p. 15 line 22f.).

p.18, line 27: "the maximum of DWR is already found at lower values of DWR" – this is confusing, possibly a typo and Lambda is meant?

No, not a typo. The maximum of DWR Ka/W of the discussed triple-frequency curves here is found at similar Lambda as for the previously discussed curves. For the discussed (MASC-derived) triple-frequency curves, however, the maximum value of DWR Ka/W is smaller and the hook shape is therefore less pronounced.

The text was untangled accordingly to avoid confusion (p. 19 line 15f.).

p.18, line 33: The "un-hooking" makes the curve behave more like the W04 and N13 curves in Figure 8. Similarly, the truncation makes the N=125 curve in Figure 9 behave more like the W04 and N13 curves as well. Not sure what to make of this.

Part of the reason for this behavior may be that both the truncation of the snowflake size distribution at smaller snowflake maximum diameters and higher values of exponential slope parameters Lambda of the snowflake size distribution lead to snowflake size distributions characterized by smaller snowflakes than the corresponding non-truncated and low-Lambda snowflake size distributions. For snowflake size distributions characterized by smaller snowflakes in general, the manuscript and previous studies (e.g., Kneifel et al. 2015) indicate that differences in (the description of) snowflake shape have a less significant effect on the corresponding triple-frequency radar signatures than for snowflake size distributions characterized by larger snowflakes.

**Response to Referee 2:**

Referee 2:
The manuscript presents a very interesting study on how additional descriptors of ice
particles can be used to better constrain a connection between scattering and physical
snowflake properties. The manuscript is well written and with exception of a few minor
problems is easy to understand. Because I would like to see the authors response to
several of my comments, I would like to suggest to publish the paper if the authors
address those concerns adequately.

Major comments:
1. The authors are modeling snowflakes as collection of solid ice spheres with a pre-
scribed mass, D and SAV. How realistic this assumption is? Leinonen and Moisseev
(2015) have argued that using spheroids (or spheres) of solid ice in place of the
crystals leads to the formation of much denser aggregates. Or in the other words is it
possible to match mass, D and SAV of a realistic snowflake using a set of spheres?
Leinonen, J., and D. Moisseev (2015), What do triple-frequency radar signa-
tures reveal about aggregate snowflakes?, J. Geophys. Res. Atmos.,
120,doi:10.1002/2014JD022072.

Authors:

The Referee raises an interesting point that deserves additional attention in the manuscript: limitations of the applied method for representing snowflakes by collections of ice spheres and implications for the presented analysis. While an exhaustive assessment of the uncertainties associated with the used approach is beyond the scope of this study, additional information has been included in the revised manuscript to outline and illustrate limitations and implication for the analysis.

The discussion of limitations of the applied method for representing snowflakes by collections of ice spheres and of implications for the presented analysis forms the new Appendix A of the revised manuscript (combined with plots S13-S15 in the Supplement). Appendix A discusses specifically the snowflake size-dependent limitation of the modeling approach based on the used density-diameter relationship (p. 25 line 16ff.); The requirement $xi^3 = Ncl =$ integer is now explicitly included in the text (p. 25 line 7ff.); And the difference between given snowflake diameter D and the actual diameters Dcl of 500 generated collections of randomly distributed ice spheres inside a spherical bounding volume of diameter D is discussed (p. 26 line 1ff.). Additionally, the conceptual difference between the applied approach of parameterizing snowflakes by prescribed effective microstructural properties and other methods that have modeled snowflakes by aggregation of many detailed constituent ice crystals (e.g., Leinonen and Moisseev 2015) is pointed out in the revised manuscript (p. 10 line 30ff.).

2. While modelling scattering from soft spheroids, the authors have used the assump-
tions that ice particles are randomly oriented. This assumption is supported by the
presented observations of orientation angles as shown in Fig. 2. However, this
assumption contradicts dual-polarization and multi-frequency radar observations, see
work of Matrosov et al for example. For example, differential reflectivity values,
Zdr, characteristic of aggregates lie in the range from 0 to 1 dB. This range can be
reproduced by soft spheroids with aspect ratio of 0.6 and a preferential horizontal
orientation. If the random orientation is assumed the expected Zdr value would 0 dB. A
possible explanation of the discrepancy is the difference in an optical and microwave
definition of particle shape. Imagine that an ice particle consists of a horizontally
aligned spheroid and an attached dendritic crystal, such that the crystal orientation
angle is different from 0. If most of the spheroid mass is much larger than that of
the dendrite than for radar scattering calculations the particle can be assumed to be
spheroidal and horizontally aligned. The shadow image of the particle would be
different from the spheroid, and the orientation angle of this complex particle is
different from 0. At the moment, we don't know what is the best assumption of a
particle shape and what is the relation between optical and microwave particle
properties of ice particles. Therefore, we should use models that covers a larger

range of possible backscattering properties. I suggest that instead of random
orientation the authors would use a spheroid with a preferential horizontal
orientation.

Randomly oriented spheroids are used to ensure consistency throughout the manuscript with regard to analyzed snowflake observations and snowflake particle models. (Characteristic) snowflakes with such extreme geometries as the example given by the Referee seem to be unlikely for realistic snowflake aggregation and riming processes, even beyond the snowflake observations that are included in the manuscript (see, e.g., Kikuchi et al. 2013 for photos and sketches of many characteristically shaped snow crystals).

Nonetheless, the calculations have also been performed for horizontally oriented spheroids. The results are shown in the following figure (= Figure 7 of the manuscript + corresponding triple-frequency radar signatures for horizontally oriented spheroids with aspect ratios of 0.6 and 0.2, indicated by thick dashed and solid cyan lines, respectively). It is seen that using horizontally oriented soft spheroids instead of randomly oriented soft spheroids does not increase the modeled range of 'spheroidal' triple-frequency radar signatures relevant to the analysis in this study; and for a moderate aspect ratio of 0.6, often found to be a reasonable average aspect ratio of snowflakes (e.g. by Matrosov et al.), the triple-frequency curve for horizontally oriented soft spheroids is very similar to the corresponding curve obtained for spheroids with random orientation indicated by the black dashed line (not dash-dotted line) in each plot.

[Figure]

These findings are in agreement with results of Leinonen et al (2012), who already analyzed triple-frequency radar signatures of various distributions of preferentially horizontally oriented soft spheroids. They also included realistic variations of other snowfall characteristics like gamma size distributions, snowflake mass-diameter relationships and aspect ratios in their study. Even when snowflake orientation and these other snowfall characteristics are varied over a wide (realistic) range, the total range of modeled triple-frequency radar signatures for soft spheroids then still does not show significantly better agreement with previous observations of snowfall triple-frequency radar signatures (e.g., Kneifel et al. 2015). Thus, even variation of several snowflake parameters for soft spheroids, including preferential snowflake

orientation, cannot explain the wide variety of snowfall triple-frequency radar signatures that were (I) observed in previous studies (e.g., indicated by rectangles in the figure above) and (II) modeled by snowflake 3D shape models in previous studies or by the Ncl ice sphere collections in this study (gray area in the figure above). Therefore, including preferentially horizontally oriented spheroids as particle model in this study would not change any of the discussed results for (randomly oriented) soft spheroids significantly and would not change any of the drawn conclusions.

With regard to the apparent contradiction to previous work by Matrosov et al. who analyzed snowfall by dual-polarization radar as indicated by the Referee: It seems that the cases which they analyzed and modeled by preferentially horizontally oriented soft spheroids were confined to snowflakes where aggregation and riming are small and to snowflakes with characteristic maximum dimensions <= 2 mm, often <= 1 mm. Such small pristine snowflakes are not fully representative of most snowfall observed and discussed in the presented manuscript, which generally includes non-planar aggregate and rimed snowflakes of various degrees of aggregation and riming and often of larger sizes (see Fig. 1 for examples).

The minor effect of including preferentially horizontally oriented spheroids for the analysis presented in this study is stated on p. 18 line 32ff in the revised manuscript.

```
3. The authors state that they use observations from 47 snowstorms observed in Utah
and 7 storms in Barrow, which resulted in 4.3 · 10⁵ and 10⁴ snowflake observations.
The number of snowflakes sounds to be too small. I would expect that 4.3 · 10⁵ would
be a good number of snowflakes recorded during a single snowstorm. Of course, this
number depends on the instrument sampling volume and how often observations are
made. Both of which are not discussed in the paper. Could you please include more
information on how the measurements are made, how PSD are computed, how often
images are taken, etc.
```

On the order of $10^2$ to $10^4$ snowflakes were recorded during each of the the analyzed snowstorms. Low values of $10^2$ correspond to weak snowstorms at Barrow where exposure to strong crosswinds also affected the overall sampling efficiency. Averaged over each snowstorm period, the measurements correspond to snowflake sampling rates of $10^2$ to $10^3$ snowflake observations per hour. These sampling rates are on the low end of the sampling rates observed with a 2D disdrometer by Brandes et al. (2007) that typically sampled $10^2$ to $10^3$ snowflakes in 5 minutes, for example. But considering the small effective sampling volume of less than 50 cm$^3$ for the high-resolution MASC measurements of individual snowflakes at a resolution of about 30 um, these sampling rates of $10^2$ to $10^3$ snowflake observations per hour are typical for the MASC measurements collected at Alta and at Barrow.

Following the Referee's request, additional information is given about the MASC instrument (p. 3 line 18ff.) and about the snowflake measurements and analysis presented in this study (p. 6 line 2ff.). For further details on the MASC, the reader is referred to Garrett et al. (2012) who introduced the MASC (p. 3 line 11f.).

```
Minor comments:
1. There is a lot of discussion about the snowflake complexity, while
the main focus of the paper on SAV. It is a little bit confusing? Could you consider
them together or explain how you compute SAV from observations?
```

Parts of the relevant text passages were rearranged and rewritten for easier understanding overall and to clarify the relation between snowflake complexity chi (obtained from 2D projection images) and normalized snowflake surface-area-to-volume ratio xi (3D microstructural parameter). They are linked in this study by assuming a similar relation to express their respective dependence on snowflake diameter D on average (given by modified power laws for xi(D) and chi(D)).

With respect to computing SAV from snowflake complexity observations: What has been done in the manuscript is to estimate xi(D) from chi(D) relationships to then model triple-frequency radar signatures. Chi is used as an indicator of xi for this purpose, not to suggest that SAV of a snowflake can be computed

from the complexity value that was obtained from the corresponding snowflake images. The derivation of xi(D) from chi(D) is explained in the description of Eq. (8): This is done by mapping chi onto xi.

The introduction of the normalized surface-area-to-volume ratio xi and the quantification of xi from synthetically generated xi(D) and from chi(D) obtained from snowflake observations now form a new section in the revised manuscript (new Section 3.2). See specifically the description of Eq. (8) in the new Section 3.2 for how to estimate xi(D) relationships from observed snowflake complexity chi(D) relationships (p. 10 line 3ff.). The use of chi as an indicator of xi for deriving snowfall triple-frequency radar signatures is now emphasized throughout the manuscript to avoid suggesting that SAV of a snowflake can simply be computed from the corresponding snowflake complexity chi obtained from the images of that snowflake; and for the MASC data analyzed in this study, more specifics are now provided on how triple-frequency radar signatures are derived from snowflake complexity observations via the chi(D) and xi(chi) relationships (p. 19 line 13ff., p. 21 line 3ff.).

```
2. In the paper, the snowflake complexity parameter is used to describe ice parti-
cle properties. On page 6. the authors mention that for particles with D>=3 mm the
complexity parameter is larger than 1, which corresponds to aggregates. What about
heavily rimed aggregates? Would the complexity parameter and SAV be different from
1? It is not directly related to this study, but I have seen other studies where this
parameter is used as an indicator of riming. I would be interesting to know, whether
this parameter can be used as a riming indicator for all types of particles,
regardless of their initial complexity.
```
Riming is a major factor in determining snowflake complexity, but different unrimed snowflakes can also be marked by different snowflake complexity values. So, riming is not the only factor that determines snowflake complexity. A simplistic quantitative statement like 'snowflake x has complexity y and therefore shows a degree or amount of riming of z' then seems unrealistic without having additional information on the snowflake microstructure or, alternatively, defining the degree or amount of riming strictly with respect to complexity.

Nonetheless, substantial riming of any snowflake is expected to lead to a coarser microstructure than the microstructure of the originally unrimed snowflake and thus to a lower snowflake complexity (but not necessarily always to a complexity value of 1, especially for large snowflakes, as this would basically suggest the growth of (spherical) graupel particles beyond diameters of more than a few mm for large initially unrimed aggregate snowflakes). Based on this general effect of riming on the snowflake microstructure, Garrett and Yuter (2014) used snowflake complexity (with a definition of complexity that additionally includes brightness variations across the snowflake images) as an indicator of riming to distinguish between several general snow types: heavily rimed graupel, rimed snowflakes, and aggregates for a wide variety of particles (without knowledge of and independent of their initial complexity before any possible riming set in).

The expected general effect of snowflake riming on snowflake complexity is now briefly mentioned in the description of chi in the revised manuscript to better illustrate the meaning of snowflake complexity (p. 4 line 2ff.).